# ON THE ROLE OF EDGE DEPENDENCY IN GRAPH GENERATIVE MODELS

## ABSTRACT

In this work, we introduce a novel evaluation framework for generative models of graphs, emphasizing the importance of model-generated graph *overlap* (Chanpuriya et al., 2021) to ensure both accuracy and edge-diversity. We delineate a hierarchy of graph generative models categorized into three levels of complexity: edge independent, node independent, and fully dependent models. This hierarchy encapsulates a wide range of prevalent methods. We derive theoretical bounds on the number of triangles and other short-length cycles producible by each level of the hierarchy, contingent on the model overlap. We provide instances demonstrating the asymptotic optimality of our bounds. Furthermore, we introduce new generative models for each of the three hierarchical levels, leveraging dense subgraph discovery (Gionis & Tsourakakis, 2015). Our evaluation, conducted on real-world datasets, focuses on assessing the output quality and overlap of our proposed models in comparison to other popular models. Our results indicate that our simple, interpretable models provide competitive baselines to popular generative models. Through this investigation, we aim to propel the advancement of graph generative models by offering a structured framework and robust evaluation metrics, thereby facilitating the development of models capable of generating accurate and edge-diverse graphs.

## 1 INTRODUCTION

Mathematicians and physicists have long studied random discrete structures, such as sets, permutations (Kolchin & Chistyakov, 1975), and graphs (Erdös & Rényi, 1960). In recent decades, random graphs (also known as graph generative models) have become increasingly important for modeling and understanding complex networks across a wide range of domains (Easley & Kleinberg, 2010). Random graph models provide a way to capture the uncertainty that is inherent in many real-world networks. For example, in the social sciences, random graphs have been used to study the spread of information and disease (Keliger et al., 2023) and the evolution of social networks (Newman et al., 2002). In economics, random graphs have been used to study the behavior of financial markets and the spread of economic shocks (Jackson et al., 2008). In physics, random graphs have been used to study the behavior of disordered materials, the spread of traffic jams, and the emergence of synchronization in complex systems (Dorogovtsev & Mendes, 2003). In drug discovery, random graphs have been used to study the interaction between drugs and proteins, the design of new drugs, and the prediction of drug toxicity (Pan et al., 2022). In bioinformatics, random graphs have been used to study the structure of DNA and proteins and brain networks (Simpson et al., 2011).

Deep learning (LeCun et al., 2015) has led to the development of new methods for fitting random graph models such as graph autoencoders (Kipf & Welling, 2016; Simonovsky & Komodakis, 2018), NetGAN (Bojchevski et al., 2018), GraphGAN (Wang et al., 2018), GraphRNN (You et al., 2018), among many others. Such methods are typically evaluated based on two key dimensions: efficiency and output quality. Output quality measures how closely the generated graphs resemble real-world graphs that they aim to model. A common way to measure output quality is to measure a model's tendency to produce graphs that have similar statistical properties to a target input graph or set of graphs. Common examples of statistics that are considered include the average degree, maximum degree, degree assortativity, motif counts, and shortest-path measures. In terms of motif counts triangles play a major role. This is due to the fact that triangle density is a hallmark property of

several types of networks, especially social networks, and plays an important role in network analysis, e.g., Watts & Strogatz (1998); Yin et al. (2017); Tsourakakis et al. (2017).

Despite the development of the aforementioned sophisticated algorithms for fitting graph generative models, Chanpuriya et al. (2021) highlight significant limitations of any method that fits an *edge independent* random graph model, where each edge is added independently with some (non-uniform) probability. Informally (see Section 2 for details), *any* edge independent model is limited with respect to how many triangles it can generate, *unless* it memorizes a single target network. The study's findings have implications for the use of neural-network-based graph generative models, such as NetGAN (Bojchevski et al., 2018; Rendsburg et al., 2020), Graphite (Grover et al., 2019), and MolGAN (De Cao & Kipf, 2018), all of which fit edge independent models.

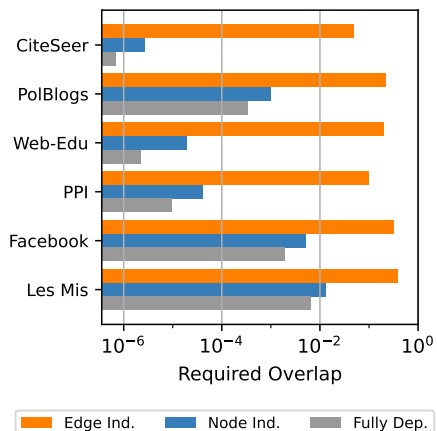

Figure 1: Minimum levels of overlap required to replicate some real world networks' triangle counts in expectation.

The results of Chanpuriya et al. (2021) introduce the notion of *overlap*, to quantify the diversity of the graphs generated by a given model. Intuitively, useful models that produce diverse output graphs have bounded overlap. Further, without the requirement of bounded overlap, it is trivial to generate graphs that have the same statistical properties as a given input graph, by simply memorizing that graph and outputting it with probability 1. Formally, Chanpuriya et al. (2021) show a trade-off between the number of triangles that an edge independent model can generate and the model overlap.

The notion of overlap will play a central role in our work. In particular, we introduce the following definition, which generalizes the concept of overlap beyond edge independent models. For simplicity, we focus on node-labeled, unweighted, and undirected graphs.

**Definition 1 (Overlap).** *Suppose $\mathcal{A}$ is a distribution over binary adjacency matrices of undirected unweighted graphs on $n$ nodes without self-loops. Let the 'volume' of $\mathcal{A}$ be the expected number of edges in a sample* $\mathrm{Vol}(\mathcal{A}) \coloneqq \mathbb{E}_{\mathbf{A} \sim \mathcal{A}} \left[ \sum_{i<j} \mathbf{A}_{ij} \right]$ . *The overlap of $\mathcal{A}$ is the expected number of shared edges between two independent samples, normalized by volume:*

$$\mathrm{Ov}(\mathcal{A}) \coloneqq \frac{\mathbb{E}_{\mathbf{A},\mathbf{A}'\sim\mathcal{A}} \left[ \sum_{i<j} \mathbf{A}_{ij} \cdot \mathbf{A}'_{ij} \right]}{\mathrm{Vol}(\mathcal{A})}.$$

**Contributions.** Our key contributions are as follows:

- We introduce a structured hierarchy of graph generative models categorized into three levels of complexity: edge independent, node independent, and fully dependent models. These levels encompass a wide array of today's prevalent methods as depicted in Table 1.

- We establish theoretical limits on the number of triangles and other short-length cycles that can be generated by each level of our hierarchy, based on the model overlap – see Table 1 for a summary. We also present instances demonstrating that our bounds are (asymptotically) optimal. Our results generalize those of Chanpuriya et al. (2021), which only considered edge independent models: we show that models at higher levels of our hierarchy can achieve higher triangle density given a fixed level of overlap. This finding matches empirical observations that such models can replicate triangle counts of a network with a much lower edge overlap – see Figure 1.

- We introduce new generative models for each of the three levels, grounded on dense subgraph discovery (Gionis & Tsourakakis, 2015) through maximal clique detection (Eppstein et al., 2010). We assess our models alongside other popular models on real-world datasets, with a focus on output quality and the overlap. We observe that our simple models are comparable to deep models in aligning graph statistics at various levels of overlap, though all models generally struggle at this task at lower overlap.

| Model | Upper Bound on $\triangle/n^3$ | Examples |
|---|---|---|
| Edge Independent | $\mathrm{Ov}(\mathcal{A})^3$ | Erdős–Rényi, SBM, NetGAN |
| Node Independent | $\mathrm{Ov}(\mathcal{A})^{3/2}$ | Variational Graph Auto-Encoder (VGAE) |
| Fully Dependent | $\mathrm{Ov}(\mathcal{A})$ | GraphVAE, GraphVAE-MM, ERGM |

Table 1: Preview of results. The level of edge dependency in graph generative models inherently limits the range of graph statistics, such as triangle counts, that they can produce. Note that $\mathrm{Ov}(\mathcal{A}) \in [0, 1]$, so a higher power on $\mathrm{Ov}(\mathcal{A})$ means a tighter bound on the number of triangles.

## 2 RELATED WORK

Random graph theory has a long history dating back to the seminal works of Gilbert (1959) and Erdős & Rényi (1960). The $G(n, p)$ Erdős-Rényi model is a model of a random graph in which each possible edge on $n$ labeled nodes is present with a given probability $p$. For convenience we set the vertex set $V = [n] = \{1, \ldots, n\}$. As in the $G(n, p)$ model, we focus in this work on the set of all possible $2^{\binom{n}{2}}$ *labeled graphs*. In recent decades, random graphs have become a crucial area of study. A wealth of random graph models that mimic properties of real world networks have been developed, including the Barabási-Albert-Bollobas-Riordan model (Albert & Barabási, 2002; Bollobás et al., 2001), the Watts-Strogatz model (Watts & Strogatz, 1998), Kronecker graphs (Leskovec et al., 2010a), and the Chung-Lu model (Chung et al., 2006).

An important class of random graph models consider nodes with latent embeddings in $\mathbb{R}^d$ and set the probability of an edge between each pair $u, v$ to be some function of the inner product of their embeddings (Young & Scheinerman, 2007; Athreya et al., 2017). Significant work has studied neural-network based embedding methods such as DeepWalk (Perozzi et al., 2014), node2vec (Grover & Leskovec, 2016), VERSE (Tsitsulin et al., 2018), and NetMF (Qiu et al., 2018). Seshadhri et al. (2020) proved that even when the probability function is non-linear, certain types of models based on low-dimensional embeddings cannot simultaneously encode edge sparsity and high triangle density, a hallmark property of real-world networks (Easley & Kleinberg, 2010; Tsourakakis, 2008).

In recent years, there has been a growing interest in deep graph generation methods (Zhu et al., 2022), such as graph autoencoders (Kipf & Welling, 2016), GraphGAN (Wang et al., 2018), GraphRNNs (You et al., 2018), MolGAN (De Cao & Kipf, 2018), GraphVAE (Simonovsky & Komodakis, 2018), NetGAN (Bojchevski et al., 2018), and CELL (Rendsburg et al., 2020).

Chanpuriya et al. (2021) considered the class of edge independent models, namely all generative models that toss independent coins for each possible edge. This class of models includes both classic models such as the $G(n, p)$ and modern methods such as NetGAN. These models can be encoded by a symmetric probability matrix $\boldsymbol{P} \in [0, 1]^{n \times n}$, where $\boldsymbol{P}_{ij}$ is the probability that nodes $i, j$ are connected. Chanpuriya et al. defined the overlap as the expected fraction of common edges between two sampled graphs from $\boldsymbol{P}$. Their key result is that *any* edge independent model with bounded overlap cannot have too many triangles in expectation. However, many generative models are not edge independent and therefore their results do not provide any insights into how many triangles they can generate for a bounded overlap. For example, exponential random graph models (ERGMs) (Frank & Strauss, 1986) go beyond edge independent models as they model the probability of observing a specific network as a function of the network's features, such as the number of edges, triangles, and other network motifs (e.g., stars and cycles). This allows for modeling higher order dependencies at the cost of learning efficiency (Chatterjee & Diaconis, 2013).

## 3 HIERARCHY OF GRAPH GENERATIVE MODELS

Our main conceptual contribution is a hierarchical categorization of graph generative models into three nested levels: edge independent, node independent, and fully dependent. For each level, we prove upper bounds on triangle count with respect to overlap; we defer the proof of these bounds to Section 4, and generalized bounds on $k$-cycles to Appendix A.2. Also for each level, we provide a simple characteristic family of graphs that achieves this upper bound on triangle counts, showing that the bound is asymptotically tight. We also discuss examples of prior models that fit into each level.

## 3.1 EDGE INDEPENDENT (EI) MODEL

We begin with the well-established class of edge independent models (Chanpuriya et al., 2021). This class includes many important models, such as the classical Erdős-Rényi and stochastic block models (SBM) (Abbe, 2017), along with modern deep-learning based models such as NetGAN (Bojchevski et al., 2018; Rendsburg et al., 2020), Graphite (Grover et al., 2019), and MolGAN (De Cao & Kipf, 2018) as they ultimately output a probability matrix based on which edges are sampled independently.

**Definition 2** (Edge Independent Graph Model). *An edge independent (EI) graph generative model is a distribution $\mathcal{A}$ over symmetric adjacency matrices $\boldsymbol{A} \in \{0,1\}^{n \times n}$, such that, for some symmetric $\boldsymbol{P} \in [0,1]^{n \times n}$, $\boldsymbol{A}_{ij} = \boldsymbol{A}_{ji} = 1$ with probability $\boldsymbol{P}_{ij}$ for all $i,j \in [n]$, otherwise $\boldsymbol{A}_{ij} = \boldsymbol{A}_{ji} = 0$. Furthermore, all entries (i.e., in the upper diagonal) of $\boldsymbol{A}$ are mutually independent.*

Any EI graph generative model satisfies the following theorem:

**Theorem 1.** *Any edge independent graph model $\mathcal{A}$ with overlap $\mathrm{Ov}(\mathcal{A})$ has at most $\frac{1}{6}n^3 \cdot \mathrm{Ov}(\mathcal{A})^3$ triangles in expectation. That is, for $\boldsymbol{A}$ drawn from $\mathcal{A}$, letting $\Delta(\boldsymbol{A})$ be the number of triangles in $\boldsymbol{A}$,*

$$\mathbb{E}\left[\Delta(\boldsymbol{A})\right] \leq \frac{n^3}{6} \cdot \mathrm{Ov}(\mathcal{A})^3. \tag{1}$$

Chanpuriya et al. (2021) give a spectral proof of Theorem 1. We provide an alternative proof based on an elegant generalization of Shearer's Entropy Lemma (Alon & Spencer, 2016; Chung et al., 1986) due to Friedgut (2004). Furthermore, we show that the upper bound is tight. Consider the $G(n, p)$ model. The expected volume is $\mathrm{Vol}(\mathcal{A}) = p \cdot \binom{n}{2}$, whereas the expected number of shared edges between two samples is $p^2 \cdot \binom{n}{2}$, yielding an overlap of $p$ (see Definition 1). Furthermore, since any triangle is materialized with probability $p^3$, by the linearity of expectation there are $O(n^3 \cdot p^3)$ triangles, which shows that inequality 1 is tight up to constant factors.

## 3.2 FULLY DEPENDENT (FD) MODEL

We now move to the other end of the edge dependency spectrum, namely to models that allow for arbitrary edge dependencies. A classic fully dependent model is the ERGM Frank & Strauss (1986); Chatterjee & Diaconis (2013) as it can model arbitrary higher order dependencies.

**Definition 3** (Fully Dependent Graph Model). *A fully dependent graph generative model allows for any possible distribution $\mathcal{A}$ over symmetric adjacency matrices $\boldsymbol{A} \in \{0,1\}^{n \times n}$.*

Allowing for arbitrary edge dependencies enables us to have models with significantly more triangles as a function of the overlap.

**Theorem 2.** *For any fully dependent (arbitrary) model $\mathcal{A}$ with overlap $\mathrm{Ov}(\mathcal{A})$, the expected number of triangles is at most $\frac{1}{2}n^3 \cdot \mathrm{Ov}(\mathcal{A})$.*

As in the case of EI models, there is a simple instance that shows that the bound is tight. Specifically, consider a model that outputs a complete graph with probability $p$ and an empty graph otherwise. Each edge occurs with probability $p$, so by the same computation as for EI models, the overlap is $p$. As for the triangle count, there are no triangles when the graph is empty, but when it is complete, there are all $\binom{n}{3}$ possible triangles. Thus, the expected number of triangles is $O(n^3 \cdot p) = O(n^3 \cdot \mathrm{Ov}(\mathcal{A}))$, again showing that our bound is tight up to constant factors

At a high level, fully dependent graph generative models often arise when methods sample a graph-level embedding, then produce a graph sample from this embedding, allowing for arbitrary dependencies between edges in the sample. A specific example is the graph variational auto-encoder (GraphVAE) (Simonovsky & Komodakis, 2018), in which decoding involves sampling a single graph-level embedding $\boldsymbol{x_G} \in \mathbb{R}^k$, and the presence of each edge is an independent Bernoulli random variable with a parameter that is some function $f_{ij}$ of $\boldsymbol{x}$: $\boldsymbol{A}_{ij} = \mathrm{Bernoulli}(f_{ij}(\boldsymbol{x_G}))$. In particular, these functions are encoded by a fully-connected neural network. Assuming these $f_{ij}$ can closely approximate the sign function, with a 1-dimensional graph embedding ($k = 1$), this model can in fact match the triangle bound of Theorem 2 by simulating the tight instance described above (outputting a complete graph with probability $p$ and the empty graph otherwise).

### 3.3 NODE INDEPENDENT (NI) MODEL

We next focus on the middle level of our hierarchy, between the two extremes of EI and FD generative models. This level is built upon the common concept of node embeddings that are stochastic and generated independently for each node.

**Definition 4** (Node Independent Graph Model). *A node independent (NI) graph generative model is a distribution $\mathcal{A}$ over symmetric adjacency matrices $\boldsymbol{A} \in \{0,1\}^{n \times n}$, where, for some embedding space $\mathcal{E}$, some mutually independent random variables $\boldsymbol{x_1}, \ldots, \boldsymbol{x_n} \in \mathcal{E}$, and some symmetric function $e : \mathcal{E} \times \mathcal{E} \to [0,1]$, the entries of $\boldsymbol{A}$ are Bernoulli random variables $\boldsymbol{A}_{ij} = Bernoulli\,(e(\boldsymbol{x_i}, \boldsymbol{x_j}))$ that are mutually independent conditioned on $\boldsymbol{x_1}, \ldots, \boldsymbol{x_n}$.*

Interestingly, the triangle bound for this class of generative models lies in the middle of the EI and FD triangle bounds.

**Theorem 3.** *Any node independent graph model $\mathcal{A}$ with overlap $\mathrm{Ov}(\mathcal{A})$ has at most $\frac{1}{6}n^3 \cdot \mathrm{Ov}(\mathcal{A})^{3/2}$ triangles in expectation.*

The proof of Theorem 3 requires expressing the probability of edges and triangles appearing as integrals in the space of node embeddings. Then we apply a continuous version of Shearer's inequality. We show that the bound of Theorem 3 is tight. Generate a random graph by initially starting with an empty graph comprising $n$ nodes. Subsequently, each node independently becomes "active" with a probability of $\sqrt{p}$, and any edges connecting active nodes are subsequently added into the graph. Note that for a given edge to be added, both of its endpoint nodes must be active, which occurs with probability $p$; this again yields an overlap of $p$ as with the prior tight examples. For a triangle to be added, all three of its endpoint nodes must be active, which occurs with probability $p^{3/2}$. Hence the expected number of triangles is $O(n^3 \cdot p^{3/2}) = O(n^3 \cdot \mathrm{Ov}(\mathcal{A})^{3/2})$.

The random graph can be represented using embeddings based on the provided definition. Consider a 1-dimensional embedding $\boldsymbol{x} \in \mathbb{R}^{n \times 1}$. Let $\boldsymbol{x}_i = 1$ with probability $\sqrt{p}$ and otherwise let $\boldsymbol{x}_i = 0$, independently for each $i \in [n]$. These coin tosses are made independently for each $i \in [n]$. To capture the edges between nodes, we define $e(\boldsymbol{x}_i, \boldsymbol{x}_j)$ as 1 when both arguments are 1, and 0 otherwise. This embedding-based representation precisely implements the described random graph.

Node independent graph models most commonly arise when methods independently sample $n$ node-level embeddings, then determine the presence of edges between two nodes based on some compatibility function of their embeddings. One notable example is the variational graph auto-encoder (VGAE) model (Kipf & Welling, 2016). In the decoding step of this model, a Gaussian-distributed node embedding is sampled independently for each node, and the presence of each possible edge is an independent Bernoulli random variable with a parameter that varies with the dot product of the corresponding embeddings as follows: $\boldsymbol{A}_{ij} = \mathrm{Bernoulli}(\sigma(\boldsymbol{x_i} \cdot \boldsymbol{x_j}))$. Thus, the VGAE model seamlessly fits into our node independent category.

## 4 PROOF OF MAIN RESULTS

In this section, we will prove the triangle count bounds stated in Section 3. We begin by introducing certain concepts that will be useful throughout our proofs.

### 4.1 THEORETICAL PRELIMINARIES

**Inner product space formulation of overlap.** We first state equivalent definitions for volume and overlap that will be useful for proofs. We first make the following observation:

**Observation 1** (Inner Product Space of Distributions over Vectors). *Suppose $\mathcal{U}, \mathcal{V}$ are distributions over real-valued $d$-dimensional vectors. Then the following operation defines an inner product:*

$$\langle \mathcal{U}, \mathcal{V} \rangle := \mathop{\mathbb{E}}_{\boldsymbol{u} \sim \mathcal{U}, \boldsymbol{v} \sim \mathcal{V}} [\boldsymbol{u} \cdot \boldsymbol{v}],$$

*where $\boldsymbol{u} \cdot \boldsymbol{v}$ is the standard vector dot product.*

Throughout, we deal with adjacency matrices of undirected graphs on $n$ nodes without self-loops, and distributions over such matrices. Dot products and inner products of these objects are taken

over $\binom{n}{2}$-dimensional vectorizations of entries below the diagonal. Let $\mathcal{F}$ denote the distribution that returns the adjacency matrix $\boldsymbol{F}$ of a graph which is fully connected (i.e., has all possible edges) with probability 1. Then $\mathrm{Vol}(\mathcal{A}) = \langle \mathcal{A}, \mathcal{F} \rangle$ and $\mathrm{Ov}(\mathcal{A}) = \frac{\langle \mathcal{A}, \mathcal{A} \rangle}{\mathrm{Vol}(\mathcal{A})}$.

These expressions allow us to derive the following upper bound on volume in terms of overlap, which applies to any graph generative model:

**Lemma 1.** *For a graph generative model $\mathcal{A}$ with overlap $\mathrm{Ov}(\mathcal{A})$, the expected number of edges* $\mathrm{Vol}(\mathcal{A})$ *is at most* $\binom{n}{2} \cdot \mathrm{Ov}(\mathcal{A})$.

*Proof.* By the definition of overlap and Cauchy-Schwarz,

$$\mathrm{Ov}(\mathcal{A}) = \frac{\langle \mathcal{A}, \mathcal{A} \rangle}{\langle \mathcal{A}, \mathcal{F} \rangle} \geq \frac{\langle \mathcal{A}, \mathcal{F} \rangle^2}{\langle \mathcal{A}, \mathcal{F} \rangle \cdot \langle \mathcal{F}, \mathcal{F} \rangle} = \frac{\langle \mathcal{A}, \mathcal{F} \rangle}{\langle \mathcal{F}, \mathcal{F} \rangle} = \frac{\mathrm{Vol}(\mathcal{A})}{\langle \mathcal{F}, \mathcal{F} \rangle}.$$

Since $\langle \mathcal{F}, \mathcal{F} \rangle = \langle \boldsymbol{F}, \boldsymbol{F} \rangle = \binom{n}{2}$, rearranging yields the desired inequality, $\mathrm{Vol}(\mathcal{A}) \leq \binom{n}{2} \cdot \mathrm{Ov}(\mathcal{A})$. $\square$

### 4.2 PROOFS OF TRIANGLE COUNT BOUNDS

We now prove the upper bounds on expected triangle count for edge independent, node independent, and fully dependent models appearing in Theorems 1, 2, and 3.

**Edge independent.** A proof for Theorem 1 based on expressing triangle count in terms of eigenvalues of $\boldsymbol{P}$ follows directly from Lemma 1 and Chanpuriya et al. (2021), but we present a different proof based on the following variant of Cauchy-Schwarz from Friedgut (2004):

$$\sum\nolimits_{ijk} a_{ij} b_{jk} c_{ki} \leq \sqrt{\sum\nolimits_{ij} a_{ij}^2 \sum\nolimits_{ij} b_{ij}^2 \sum\nolimits_{ij} c_{ij}^2}. \tag{2}$$

*Proof of Theorem 1.* Let $\boldsymbol{P}$ be the edge probability matrix of the edge independent model $\mathcal{A}$. Then, by the above inequality, for a sample $\boldsymbol{A}$ from $\mathcal{A}$,

$$\mathbb{E}[\Delta(\boldsymbol{A})] = \tfrac{1}{6} \sum\nolimits_{ijk} \boldsymbol{P}_{ij} \boldsymbol{P}_{jk} \boldsymbol{P}_{ki} \leq \tfrac{1}{6} \sqrt{\left( \sum\nolimits_{ij} \boldsymbol{P}_{ij}^2 \right)^3} = \tfrac{1}{6} \left( 2 \sum\nolimits_{i<j} \boldsymbol{P}_{ij}^2 \right)^{3/2}$$

$$= \tfrac{1}{6} \left( 2 \cdot \mathrm{Ov}(\mathcal{A}) \mathrm{Vol}(\mathcal{A}) \right)^{3/2} = \tfrac{\sqrt{2}}{3} \cdot \mathrm{Ov}(\mathcal{A})^{3/2} \mathrm{Vol}(\mathcal{A})^{3/2}.$$

Applying Lemma 1 yields $\mathbb{E}[\Delta(\boldsymbol{A})] \leq \frac{\sqrt{2}}{3} \binom{n}{2}^{3/2} \cdot \mathrm{Ov}(\mathcal{A})^3 \leq \frac{1}{6} n^3 \cdot \mathrm{Ov}(\mathcal{A})^3$. $\square$

**Fully dependent.** We now prove the triangle bound for arbitrary graph generative models, which follows from Lemma 1. Note that, given a random adjacency matrix $\boldsymbol{A} \in \{0,1\}^{n \times n}$, the product $\boldsymbol{A}_{ij} \boldsymbol{A}_{jk} \boldsymbol{A}_{ik}$ is an indicator random variable for the existence of a triangle between nodes $i, j, k \in [n]$.

*Proof of Theorem 2.* Let $\boldsymbol{A}$ be a sample from $\mathcal{A}$. From Lemma 1, we have

$$\sum\nolimits_{i<j} \mathbb{E}[\boldsymbol{A}_{ij}] = \mathrm{Vol}(\mathcal{A}) \leq \binom{n}{2} \cdot \mathrm{Ov}(\mathcal{A}).$$

So, for the expected number of triangles in a sample, we have:

$$\mathbb{E}[\Delta(\boldsymbol{A})] = \mathbb{E}\left[ \sum\nolimits_{i<j<k} \boldsymbol{A}_{ij} \boldsymbol{A}_{jk} \boldsymbol{A}_{ik} \right] = \sum\nolimits_{i<j<k} \mathbb{E}\left[ \boldsymbol{A}_{ij} \boldsymbol{A}_{jk} \boldsymbol{A}_{ik} \right]$$

$$\leq \sum\nolimits_{i<j<k} \mathbb{E}[\boldsymbol{A}_{ij}] \leq n \cdot \sum\nolimits_{i<j} \mathbb{E}[\boldsymbol{A}_{ij}] \leq n \cdot \binom{n}{2} \cdot \mathrm{Ov}(\mathcal{A}) \leq \tfrac{1}{2} n^3 \cdot \mathrm{Ov}(\mathcal{A}). \quad \square$$

**Node independent.** Proving Theorem 3 is more involved than the prior proofs, and requires first establishing the following lemma:

**Lemma 2.** *For a sample $\boldsymbol{A}$ from any node independent model on $n$ nodes and any three nodes $i, j, k \in [n]$, the probability that $i, j, k$ form a triangle is upper-bounded as follows:*

$$\mathbb{E}[\boldsymbol{A}_{ij} \boldsymbol{A}_{jk} \boldsymbol{A}_{ik}] \leq \sqrt{\mathbb{E}[\boldsymbol{A}_{ij}] \mathbb{E}[\boldsymbol{A}_{jk}] \mathbb{E}[\boldsymbol{A}_{ik}]}.$$

We prove Lemma 2 in Appendix A.1 by expressing the probability of edges and triangles appearing as integrals in the space of node embeddings, after which we can apply a theorem of Friedgut (2004) (also proved by Finner (1992)), which can be seen as a continuous version of Inequality 2. With Lemma 2 in place, it is straightforward to prove Theorem 3:

*Proof of Theorem 3.* Let $\boldsymbol{A}$ be a sample from a node independent model. For the expected number of triangles $\mathbb{E}[\Delta(\boldsymbol{A})]$ in the sample, we have that

$$\mathbb{E}[\Delta(\boldsymbol{A})] = \sum\nolimits_{i<j<k} \mathbb{E}[\boldsymbol{A}_{ij}\boldsymbol{A}_{ik}\boldsymbol{A}_{jk}] \leq \sum\nolimits_{i<j<k} \sqrt{\mathbb{E}[\boldsymbol{A}_{ij}] \cdot \mathbb{E}[\boldsymbol{A}_{ik}] \cdot \mathbb{E}[\boldsymbol{A}_{jk}]}$$

$$\leq \sqrt{\binom{n}{3} \cdot \sum\nolimits_{i<j<k} \mathbb{E}[\boldsymbol{A}_{ij}] \cdot \mathbb{E}[\boldsymbol{A}_{ik}] \cdot \mathbb{E}[\boldsymbol{A}_{jk}]}$$

$$\leq \sqrt{\binom{n}{3} \cdot \tfrac{1}{6}n^3 \cdot \mathrm{Ov}(\mathcal{A})^3} \leq \tfrac{1}{6}n^3 \cdot \mathrm{Ov}(\mathcal{A})^{3/2},$$

where second line follows from Cauchy-Schwarz and the third from Theorem 1 for EI models. $\square$

Note that we favor simplicity and interpretability of the final result over tightness. In particular, as our proofs reveal, the bound for EI models, and hence the one for NI models, is tighter when expressed in terms of both overlap and volume.

## 5 GENERATIVE MODELS LEVERAGING MAXIMAL-CLIQUE DETECTION

We shift our focus towards empirically evaluating the real-world trade-off between overlap and performance across several specific models on real-world networks. In this endeavor, we concentrate on graph generative models that, given an input adjacency matrix $\mathbf{A} \in \{0,1\}^{n \times n}$, yield a distribution $\mathcal{A}$ over adjacency matrices in $\{0,1\}^{n \times n}$. Typically, these distributions are desired to showcase two characteristics:

1. Samples from $\mathcal{A}$ should closely align with various graph statistics of the input $\mathbf{A}$, such as the degree distribution and triangle count.
2. $\mathcal{A}$ should exhibit low overlap, to deter the model from merely memorizing and reproducing $\mathbf{A}$.

Because there is generally some inherent tension between these two objectives for $\mathcal{A}$, it is desirable for the model to allow for *easy tuning of overlap*.

Recent graph generative models, especially those that incorporate edge dependency, often involve complex deep architectures with a large number of parameters that are trained with gradient descent. The abundance of parameters implies that these models may have the capacity to simply memorize the input graph. At the same time, the complexity of the approaches obscures the roles of each component in yielding performance (in particular, the role of edge dependency is unclear), and specifying a desired overlap with the input graph is generally not possible, short of heuristics like early stopping. Altogether, the preceding discussion inspires the following research questions:

1. Is the relaxed theoretical constraints on triangle counts and other graph statistics for graph generative models with edge dependency manifested in the modeling of real-world networks?
2. Is it possible to attain edge dependency through straightforward baselines that facilitate easy tuning of overlap?
3. Are such simplistic models capable of matching graph statistics on par with deep models across varied levels of overlap?

As a glimpse into our findings, we ascertain overall affirmative responses to all three of these inquiries.

### 5.1 PROPOSED MODELS

We introduce three graph generative models, one for each of the categories of our framework: EI, NI, and FD. These baseline models mainly exploit the dense subgraph structure of the input graph, specifically the set of maximal cliques (MCs) (Eppstein et al., 2010). We refer to our models as MCEI,

MCNI, and MCFD, respectively. The high-level concept of these models is to start with an empty graph and plant edges from each of the input graph $G_i$'s max cliques with some fixed probability hyperparameter $p$. How the edges are planted depends on the desired type of edge dependency and reflects the characteristic 'tight' examples for each category of the hierarchy from Section 3 – see Algorithm 1 for details.

Note that, the lower $p$, the fewer the expected edges in the final sampled graph $G_p$. To compensate for this 'residual' with respect to the input graph $G_i$, we also sample a second graph $G_r$ and return the union of $G_p$ and $G_r$. Specifically, we produce $G_r$ by sampling from a simple edge independent model that is configured so that the node degrees of the union of $G_p$ and $G_r$ match those of $G_i$ in expectation. We defer details of this auxiliary EI model to Section B. The end result is that, in each of our models, ranging the hyperparameter $p$ from 0 to 1 ranges the overlap of the resulting distribution from very low to 1. In particular, the distribution $\mathcal{A}$ goes from being nearly agnostic to the input $\mathbf{A}$ (depending only on its degree sequence), to exactly returning $\mathbf{A}$ with probability 1.

---

**Algorithm 1:** Sampling $G_p$ for our max clique-based graph generative models

---

**input** : input graph $G_i$, planting probability $p$, model type (MCEI, MCFD, or MCNI)
**output** : sampled graph $G_p$

1   initialize the sampled graph $G_p$ to be empty
2   **foreach** max clique $M \in$ input graph $G_i$ **do**
3      **if** model type is MCEI **then**
4         **foreach** edge $e \in M$ **do**
5             with probability $p$, add $e$ to $G_p$
6      **else if** model type is MCFD **then**
7         with probability $p$, add all edges in $M$ to $G_p$
8      **else if** model type is MCNI **then**
9         **foreach** node $v \in M$ **do**
10            with probability $\sqrt{p}$, set $v$ to be 'active' in $M$
11         add edges between all pairs of nodes active in $M$ to $G_p$
12   **return** $G_p$

---

## 5.2   EXPERIMENTAL SETUP AND RESULTS

**Setup.** We perform an evaluation of different graph generative models under the perspective of the *overlap* criterion. Specifically, we see how these models match different statistics of an input graph as overlap is varied. We choose 8 statistics that capture both the connectivity of a graph, as well local patterns within it. We evaluate six models in total: the three models we introduce in Section 5.1, as well as three deep learning-based models, CELL (Rendsburg et al., 2020), VGAE (Kipf & Welling, 2016), and GraphVAE (Simonovsky & Komodakis, 2018), which are EI, NI, and FD, respectively. For our methods, we vary overlap by simply varying the $p$ hyperparameter; for the other methods, we use early stopping or vary the representation dimensionality. Further details of the experimental setup are included in Appendix C. We evaluate on 8 real-world networks and 1 synthetic network. In Figure 2, we include results for LES MISERABLES (Knuth, 1993), a co-appearance network of characters in the eponymous novel; the remaining results are deferred to Appendix D.

**Results.** Our findings highlight the importance of *overlap* as a third dimension in the evaluation of graph generative models. We see that deep-learning based models, like GraphVAE and CELL, can almost fit the input graph as we allow the training to be performed for a sufficient number of epochs. However, when one wants to generate a diverse set of graphs, these models fail to match certain statistics of the input graph. For example, we see in Figure 2 that the CELL method generates graphs with a low number of triangles for the LES MISERABLES dataset when the overlap between the generated graphs is small. We observe similar results for almost all datasets and other statistics, especially those pertaining to small dense subgraphs (like 4-cliques and 4-cycles). Similarly, the GraphVAE method almost always fits the input graph in a high overlap regime. Although GraphVAE possesses greater theoretical capacity as an FD generative model, we observe that instances with low

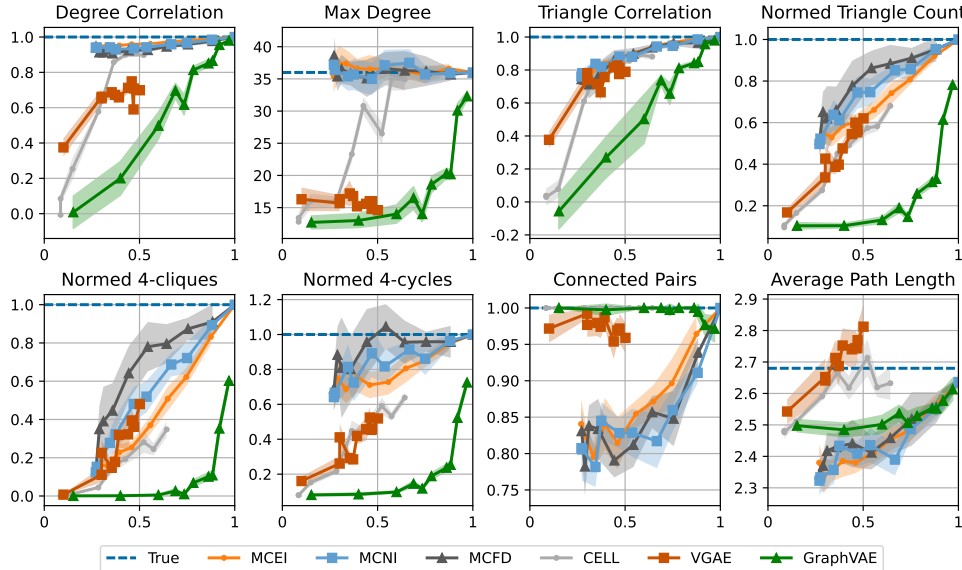

Figure 2: Statistics for LES MISERABLES as a function of the *overlap*: max degree; Pearson correlation coefficient of degree and triangle sequences; normalized triangle, 4-clique, and 4-cycle counts; fraction of connected pairs, $\sum_i \binom{|C_i|}{2} / \binom{n}{2}$ where $C_i$ is the $i^{\text{th}}$ connected component; and characteristic (average) path length. We plot the mean and 90% confidence interval across 10 trials.

overlap deviate significantly from the statistical characteristics of the input graph, as illustrated in Figure 2. Consequently, this approach fails to achieve generalization when trained on a single graph instance. The VGAE model encounters additional limitations due to its dot-product kernel, which has been demonstrated to have certain constraints in generating graphs with sparsity and triangle density (Seshadhri et al., 2020; Chanpuriya et al., 2020). As depicted in Figures 2, we observe that this model fails to adequately capture the characteristics of the input graph, even when trained for many epochs.

Surprisingly, despite their simplicity, the three introduced models exhibit desirable characteristics. Firstly, the overlap can be easily adjusted by increasing the probability hyperparameter $p$, providing more predictability compared to other models (refer to the results for the POLBLOGS dataset in the Appendix). Secondly, these models generally generate a higher number of triangles, which closely aligns with the input graph, in comparison to methods such as CELL. For instance, in Figure 2, these baselines produce graphs with a greater number of triangles, 4-cliques, and 4-cycles than CELL and GraphVAE, particularly when the overlap is low. Furthermore, as the level of dependency rises (from edge independent to fully dependent), we observe a higher triangle count for a fixed overlap. This finding supports our theoretical assertions from Section 3 regarding the efficacy of different models within the introduced hierarchy. However, a drawback of these two-stage methods is their inability to capture the connectivity patterns of the input graph, as evident from the fraction of connected pairs and the characteristic path length statistics.

## 6 CONCLUSION

We have proved tight trade-offs for graph generative models between their ability to produce networks that match the high triangle densities of real-world graphs, and their ability to achieve low *overlap* and generate a diverse set of networks. We show that as the models are allowed higher levels of edge dependency, they are able to achieve higher triangle counts with lower overlap, and we formalize this finding by introducing a three-level hierarchy of edge dependency. An interesting future direction is to refine this hierarchy to be finer-grained, and also to investigate the roles of embedding length and complexity of the embedding distributions. We also emphasize our introduction of *overlap* as a third dimension along which to evaluate graph generative models, together with output quality and efficiency. We believe these directions can provide a solid groundwork for the systematic theoretical and empirical study of graph generative models.

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

## A  DEFERRED PROOFS AND THEORETICAL RESULTS

### A.1  PROOF OF LEMMA 2

We prove Lemma 2 using the following theorem from Friedgut (2004) (see also Finner (1992)), which can be seen as a continuous version of Inequality 2:

**Theorem 4** (Friedgut (2004)). *Let $X, Y, Z$ be three independent probability spaces and let*

$$f : X \times Y \to \mathbb{R}, \quad g : Y \times Z \to \mathbb{R}, \quad and \quad h : Z \times X \to \mathbb{R}$$

*be functions that are square-integrable with respect to the relevant product measures. Then*

$$\int f(x,y)g(y,z)h(z,x)\,dx\,dy\,dz \leq \sqrt{\int f^2(x,y)\,dx\,dy \int g^2(y,z)\,dy\,dz \int h^2(z,x)\,dz\,dx}.$$

*Proof of Lemma 2.* Consider any set of three distinct nodes $(i, j, k)$. Then, by assumption, the embeddings of these nodes $Z_i, Z_j, Z_k$ are independent random variables. Let $\rho_{Z_i}, \rho_{Z_j}, \rho_{Z_k}$ be the PDFs of the respective nodes' embeddings. Recall that there exists a symmetric function $e : \mathbb{R}^k \times \mathbb{R}^k \to [0, 1]$ of two nodes' embeddings that determines the probability of an edge between them. Based on this, the probability that nodes $i, j, k$ form a triangle is:

$$\mathbb{E}[\boldsymbol{A}_{ij}\boldsymbol{A}_{jk}\boldsymbol{A}_{ik}] = \int \rho_{Z_i}(z_i)\rho_{Z_j}(z_j)\rho_{Z_k}(z_k)e(z_i, z_j)e(z_j, z_k)e(z_i, z_k)\,dz_i\,dz_j\,dz_k$$

Now, define $f(z_i, z_j) = \sqrt{\rho_{Z_i}(z_i) \cdot \rho_{Z_j}(z_j)} \cdot e(z_i, z_j)$, $g(z_j, z_k) = \sqrt{\rho_{Z_j}(z_j) \cdot \rho_{Z_k}(z_k)} \cdot e(z_j, z_k)$, and $h(z_i, z_k) = \sqrt{\rho_{Z_i}(z_i) \cdot \rho_{Z_k}(z_k)} \cdot e(z_i, z_k)$, so that

$$\mathbb{E}[\boldsymbol{A}_{ij}\boldsymbol{A}_{jk}\boldsymbol{A}_{ik}] = \int f(z_i, z_j)g(z_j, z_k)h(z_i, z_k)\,dz_i\,dz_j\,dz_k$$

Now, we can apply Theorem 4, yielding:

$$\mathbb{E}[\boldsymbol{A}_{ij}\boldsymbol{A}_{jk}\boldsymbol{A}_{ik}] \leq \sqrt{\int f^2(z_i, z_j)\,dz_i\,dz_j \int g^2(z_j, z_k)\,dz_j\,dz_k \int h^2(z_i, z_k)\,dz_i\,dz_k}$$

$$= \sqrt{\int \rho_{Z_i}(z_i)\rho_{Z_j}(z_j)e^2(z_i, z_j)\,dz_i\,dz_j \int \rho_{Z_j}(z_j)\rho_{Z_k}(z_k)e^2(z_j, z_k)\,dz_j\,dz_k \int \rho_{Z_i}(z_i)\rho_{Z_k}(z_k)e^2(z_i, z_k)\,dz_i\,dz_k}$$

$$\leq \sqrt{\int \rho_{Z_i}(z_i)\rho_{Z_j}(z_j)e(z_i, z_j)\,dz_i\,z_j \int \rho_{Z_j}(z_j)\rho_{Z_k}(z_k)e(z_j, z_k)\,dz_j\,dz_k \int \rho_{Z_i}(z_i)\rho_{Z_k}(z_k)e(z_i, z_k)\,dz_i\,dz_k}.$$

where the last inequality simply uses the fact that since the image of the function $e$ is $[0, 1]$, it must be that $e^2(x, y) \leq e(x, y)$. Finally, the lemma follows from observing that

$$\int \rho_{Z_i}(z_i)\rho_{Z_j}(z_j)e(z_i, z_j)\,dz_i\,dz_j = \mathbb{E}[\boldsymbol{A}_{ij}]. \qquad \square$$

### A.2  GENERALIZED BOUNDS FOR SQUARES AND OTHER $k$-CYCLES

We can extend the prior bounds on triangles to bounds on the expected number of $k$-cycles in graphs sampled from the generative model $\mathcal{A}$ in terms of $\mathrm{Ov}(\mathcal{A})$. For the adjacency matrix $\boldsymbol{A}$ of a graph $G$, let $C_k(\boldsymbol{A})$ denote the number of $k$-cycles in $G$. We can prove the following bounds, where the random graph models from Section 3 also provide tight examples for any generalized $k$-cycles.

**Theorem 5** (Bound on Expected $k$-cycles). *Let $\boldsymbol{A}$ be an adjacency matrix sampled from a graph generative model $\mathcal{A}$, and let $C_k(\boldsymbol{A})$ denote the number of $k$-cycles in the graph corresponding to $\boldsymbol{A}$. If $\mathcal{A}$ is edge independent, node independent, or fully dependent then $\mathbb{E}[C_k(\boldsymbol{A})]$ is bounded above asymptotically by $n^k \cdot \mathrm{Ov}(\mathcal{A})^k$, $n^k \cdot \mathrm{Ov}(\mathcal{A})^{k/2}$, and $n^k \cdot \mathrm{Ov}(\mathcal{A})$, respectively.*

*Proof.* For notational simplicity, we focus on $k = 4$. The proof directly extends to general $k$. Let $C_4(G)$ be the number of non-backtracking 4-cycles in $G$ (i.e. squares), which can be written as

$$\mathbb{E}_{\boldsymbol{A} \sim \mathcal{A}}[C_4(\boldsymbol{A})] = \frac{1}{8} \cdot \sum_{i=1}^{n} \sum_{j \in [n] \setminus \{i\}} \sum_{k \in [n] \setminus \{i,j\}} \sum_{\ell \in [n] \setminus \{i,j,k\}} \boldsymbol{A}_{ij} \boldsymbol{A}_{jk} \boldsymbol{A}_{k\ell} \boldsymbol{A}_{\ell i}.$$

The $1/8$ factor accounts for the fact that in the sum, each square is counted 8 times – once for each potential starting vector $i$ and once of each direction it may be traversed. For general $k$-cycles this factor would be $\frac{1}{2k}$. We then can bound

$$\mathbb{E}_{\boldsymbol{A} \sim \mathcal{A}}[C_4(\boldsymbol{A})] \leq \frac{1}{8} \cdot \sum_{i \in [n]} \sum_{j \in [n]} \sum_{k \in [n]} \sum_{\ell \in [n]} \boldsymbol{A}_{ij} \boldsymbol{A}_{jk} \boldsymbol{A}_{k\ell} \boldsymbol{A}_{\ell i}.$$

If $\mathcal{A}$ is an edge independent model, the bound on the expected number of 4-cycles proceeds like the one for triangles, except using the following variant of Cauchy-Schwarz:

$$\sum\nolimits_{ijkl} a_{ij} b_{jk} c_{kl} d_{li} \leq \sqrt{\sum\nolimits_{ij} a_{ij}^2 \sum\nolimits_{ij} b_{ij}^2 \sum\nolimits_{ij} c_{ij}^2 \sum\nolimits_{ij} d_{ij}^2},$$

which, letting $\boldsymbol{P} = \mathbb{E}[\boldsymbol{A}]$ be the edge probability matrix for $\boldsymbol{A} \sim \mathcal{A}$, yields

$$\begin{aligned} \mathbb{E}[C_4(\boldsymbol{A})] &= \tfrac{1}{8} \sum\nolimits_{ijkl} \boldsymbol{P}_{ij} \boldsymbol{P}_{jk} \boldsymbol{P}_{kl} \boldsymbol{P}_{li} \leq \tfrac{1}{8} \sqrt{\left(\sum\nolimits_{ij} \boldsymbol{P}_{ij}^2\right)^4} \\ &= \tfrac{1}{8} \left(2 \cdot \mathrm{Ov}(\mathcal{A}) \mathrm{Vol}(\mathcal{A})\right)^{4/2} = O\left(n^4 \mathrm{Ov}(\mathcal{A})^4\right), \end{aligned}$$

where again the last step is applying Lemma 1.

If $\mathcal{A}$ is a fully dependent model, the proof of the bound carries through almost exactly:

$$\begin{aligned} \mathbb{E}[C_4(\boldsymbol{A})] &= \sum_{i<j<k<l} \mathbb{E}[\boldsymbol{A}_{ij} \boldsymbol{A}_{jk} \boldsymbol{A}_{kl} \boldsymbol{A}_{li}] \leq \sum_{i<j<k<l} \mathbb{E}[\boldsymbol{A}_{ij}] \\ &= O(n^2) \sum\nolimits_{i<j} \mathbb{E}[\boldsymbol{A}_{ij}] = O(n^2) \cdot \mathrm{Vol}(\mathcal{A}) \leq O(n^4) \cdot \mathrm{Ov}(\mathcal{A}). \qquad \square \end{aligned}$$

If $\mathcal{A}$ is a node independent model, the bound on the expected number of 4-cycles follows as before, except using the following extended version of Theorem 4 (that we prove in Sec. A.3):

$$\int f(x,y) g(y,z) h(z,w) l(w,x) \, dx \, dy \, dz \, dw$$

$$\leq \sqrt{\int f^2(x,y) \, dx \, dy \int g^2(y,z) \, dy \, dz \int h^2(z,w) \, dz \, dw \int l^2(w,x) \, dw \, dx}.$$

Applying this equation as before yields

$$\mathbb{E}[\boldsymbol{A}_{ij} \boldsymbol{A}_{jk} \boldsymbol{A}_{kl} \boldsymbol{A}_{li}] \leq \sqrt{\mathbb{E}[\boldsymbol{A}_{ij}] \mathbb{E}[\boldsymbol{A}_{jk}] \mathbb{E}[\boldsymbol{A}_{kl}] \mathbb{E}[\boldsymbol{A}_{li}]},$$

which allows us to apply the edge independent bound for squares:

$$\begin{aligned} \mathbb{E}[C_4(\boldsymbol{A})] &= \sum_{i<j<k<l} \mathbb{E}[\boldsymbol{A}_{ij} \boldsymbol{A}_{jk} \boldsymbol{A}_{kl} \boldsymbol{A}_{li}] \leq \sum_{i<j<k<l} \sqrt{\mathbb{E}[\boldsymbol{A}_{ij}] \mathbb{E}[\boldsymbol{A}_{jk}] \mathbb{E}[\boldsymbol{A}_{kl}] \mathbb{E}[\boldsymbol{A}_{li}]} \\ &\leq \sqrt{\binom{n}{4} \sum\nolimits_{i<j<k<l} \mathbb{E}[\boldsymbol{A}_{ij}] \mathbb{E}[\boldsymbol{A}_{jk}] \mathbb{E}[\boldsymbol{A}_{kl}] \mathbb{E}[\boldsymbol{A}_{li}]} \\ &\leq \sqrt{O\left((n^4)^2 \cdot \mathrm{Ov}(\mathcal{A})^4\right)} = O(n^4 \mathrm{Ov}(\mathcal{A})^{4/2}). \end{aligned}$$

### A.3 INTEGRAL INEQUALITY AND PROOF

Here we prove the following theorem that was referenced in Section 4.2:

**Theorem 6.** *For square integrable functions $f_1, \ldots f_k$, we have:*

$$\int_{x_1} \int_{x_2} \ldots \int_{x_k} f_1(x_1, x_2) \cdot f_2(x_2, x_3) \cdot \ldots \cdot f_k(x_k, x_1) \, dx_1 \ldots dx_k$$

$$\leq \sqrt{\int_{x_1} \int_{x_2} f_1^2(x_1, x_2) dx_1 dx_2 \cdot \ldots \cdot \int_{x_k} \int_{x_1} f_k^2(x_k, x_1) dx_k dx_1}$$

Before we prove this theorem, we first prove an intermediate result that will be useful:

**Lemma 3.** *For $k \geq 3$, we have:*

$$\int_{x_1} \int_{x_2} \left( \int_{x_3} \ldots \int_{x_k} f_2(x_2, x_3) \cdot \ldots \cdot f_k(x_k, x_1) dx_3 \ldots dx_k \right)^2 dx_1 dx_2$$

$$\leq \int_{x_2} \int_{x_3} f_2^2(x_2, x_3) dx_2 dx_3 \cdot \ldots \cdot \int_{x_k} \int_{x_1} f_k^2(x_k, x_1) dx_k dx_1$$

*Proof.* We proceed using induction. For $k = 3$, we have

$$\int_{x_1} \int_{x_2} \left( \int_{x_3} f_2(x_2, x_3) \cdot f_3(x_3, x_1) dx_3 \right)^2 dx_1 dx_2$$

$$\leq \int_{x_1} \int_{x_2} \int_{x_3} f_2^2(x_2, x_3) dx_3 \int_{x_3} f_3^2(x_3, x_1) dx_3 dx_2 dx_1$$

$$= \int_{x_2} \int_{x_3} f_2^2(x_2, x_3) dx_2 dx_3 \cdot \int_{x_1} \int_{x_3} f_3^2(x_3, x_1) dx_3 dx_1,$$

where the first inequality follows from Cauchy-Schwarz.
Now assume the statement is true for some $k > 3$. We will prove the induction step for $k + 1$:

$$\int_{x_1} \int_{x_2} \left( \int_{x_3} \ldots \int_{x_{k+1}} f_2(x_2, x_3) \cdot \ldots \cdot f_{k+1}(x_{k+1}, x_1) dx_3 \ldots dx_{k+1} \right)^2 dx_1 dx_2$$

$$= \int_{x_1} \int_{x_2} \left( \int_{x_3} f_2(x_2, x_3) \left( \int_{x_4} \int_{x_{k+1}} f_2(x_2, x_3) \cdot \ldots \cdot f_{k+1}(x_{k+1}, x_1) \ldots dx_{k+1} \right) dx_3 \right)^2 dx_1 dx_2$$

$$\leq \int_{x_2} \int_{x_3} f_2^2(x_2, x_3) dx_3 dx_2 \int_{x_1} \int_{x_3} \left( \int_{x_4} \ldots \int_{x_{k+1}} f_3(x_3, x_4) \ldots f_{k+1}(x_{k+1}, x_1) dx_4 \ldots dx_{k+1} \right)^2 dx_3 dx_1$$

$$\leq \int_{x_2} \int_{x_3} f_2^2(x_2, x_3) dx_2 dx_3 \cdot \ldots \cdot \int_{x_{k+1}} \int_{x_1} f_{k+1}^2(x_{k+1}, x_1) dx_{k+1} dx_1,$$

where the first inequality follows from Cauchy-Schwarz, and the last one from the inductive hypothesis. □

Now, we are ready to prove Theorem 6.

*Proof.*

$$\int_{x_1} \int_{x_2} \ldots \int_{x_k} f_1(x_1, x_2) \cdot f_2(x_2, x_3) \cdot \ldots \cdot f_k(x_k, x_1) \, dx_1 \ldots dx_k$$

$$= \int_{x_1} \int_{x_2} f_1(x_1, x_2) \left( \int_{x_3} \ldots \int_{x_k} f_2(x_2, x_3) \cdot \ldots \cdot f_k(x_k, x_1) dx_3 \ldots dx_k \right) dx_2 dx_1$$

$$\leq \sqrt{\int_{x_1} \int_{x_2} f_1^2(x_1, x_2) dx_1 dx_2 \int_{x_1} \int_{x_2} \left( \int_{x_3} \ldots \int_{x_k} f_2(x_2, x_3) \cdot \ldots \cdot f_k(x_k, x_1) dx_3 \ldots dx_k \right)^2 dx_1 dx_2}$$

$$\leq \sqrt{\int_{x_1} \int_{x_2} f_1^2(x_1, x_2) dx_1 dx_2 \cdot \ldots \cdot \int_{x_k} \int_{x_1} f_k^2(x_k, x_1) dx_k dx_1},$$

where the first inequality follow from Cauchy-Schwarz and the second from Lemma 3. □

# B  FURTHER DETAILS ON OUR BASELINE GRAPH GENERATIVE MODELS

We provide additional details on our new graph generative models that were introduced in Section 5.1. Recall that our models are based on sampling a primary graph $G_p$ based on the maximal cliques of the input graph $G_i$, then a 'residual' graph $G_r$ based on the node degrees of $G_i$, and finally returning the union of $G_p$ and $G_r$. Specifically, we produce $G_r$ by sampling from a simple edge independent model, the *odds product model* (Hoff, 2003; De Bie, 2011; Chanpuriya et al., 2021). This is as a variant of the well-known Chung-Lu configuration model (Aiello et al., 2001; Chung & Lu, 2002a;b), which produces graphs that match an input degree sequence in expectation. The odds product model does the same, except without constraints on the input degree sequence. In this work, we generalize this model to produce a sampled graph $G_r$ such that its union with a $G_p$ matches the input graph's degree distribution in expectation. Here we discuss some more details about fitting and sampling from the odds product model to produce $G_r$.

In the odds product model, there is a logit $\ell \in \mathbb{R}$ assigned to each node, and the probability of an edge occurring between two nodes $i$ and $j$ rises with the sum of their logits, and is given by $\sigma(\ell_i + \ell_j)$, where $\sigma$ is the logistic function. To fit this model, we find a vector $\ell \in \mathbb{R}^n$ of $n$ logits, one for each node, such that a sample from the model has the same expected node degrees (i.e., row and column sums) as the original graph. This model is edge independent, so to use it, we simply find the expected adjacency matrix of edge probabilities, then sample entries independently to produce a graph.

We use largely the same algorithm as Chanpuriya et al. (2021) to fit the model, but with some modifications, since our goal is slightly more complex. Recall that in the context of this work, we actually sample from this model to produce a 'residual' graph with adjacency matrix $A_r$; we then return the union $A_u$ of this graph with the primary sampled graph $A_p$, with the goal that, in expectation, $A_u$ has the same node degrees as the input graph $A_i$. (Note that this is a strict generalization of the same problem without the primary sampled graph, which is apparent from setting $A_p$ to be all zeros.) Denote the degree sequence of the input graph and the expected degree sequence of the union graph by $d = A_i \mathbf{1} \in \mathbb{R}^n$ and $\hat{d} = \mathbb{E}[A_u]\mathbf{1} \in \mathbb{R}^n$, respectively. Fitting the model is a root-finding problem: we seek $\ell \in \mathbb{R}^n$ such that the degree errors are zero, that is, $\hat{d} - d = \mathbf{0}$. We employ the multivariate Newton-Raphson method to find the root, for which we must calculate the Jacobian matrix $J$ of derivatives of the degree errors with respect to the entries of $\ell$. The following calculation largely mirrors that of Chanpuriya et al. (2021), though we must account for the union of graphs. If an edge is planted in the primary sample with probability $p$ and in the residual sample with probability $q$, then it exists in the union with probability $1 - (1 - p)(1 - q)$. In our notation,

$$\mathbb{E}[A_{u\,ij}] = 1 - \big(1 - \mathbb{E}[A_{p\,ij}]\big) \cdot \big(1 - \mathbb{E}[A_{r\,ij}]\big)$$
$$= 1 - \big(1 - \mathbb{E}[A_{p\,ij}]\big) \cdot \big(1 - \sigma(\ell_i + \ell_j)\big).$$

Letting $\delta_{ij}$ be 1 if $i = j$ and 0 otherwise (i.e. the Kronecker delta), we have

$$
\begin{aligned}
\frac{\partial \hat{d}_i}{\partial \ell_j} &= \frac{\partial}{\partial \ell_j} \sum_{k \in [n]} \mathbb{E}[A_{u\,ik}] \\
&= \frac{\partial}{\partial \ell_j} \sum_{k \in [n]} \big(1 - \big(1 - \mathbb{E}[A_{p\,ik}]\big) \cdot \big(1 - \sigma(\ell_i + \ell_k)\big)\big) \\
&= \frac{\partial}{\partial \ell_j} \big(1 - \big(1 - \mathbb{E}[A_{p\,ij}]\big) \cdot \big(1 - \sigma(\ell_i + \ell_j)\big)\big) \\
&\quad + \delta_{ij} \sum_{k \in [n]} \frac{\partial}{\partial \ell_i} \big(1 - \big(1 - \mathbb{E}[A_{p\,ik}]\big) \cdot \big(1 - \sigma(\ell_i + \ell_k)\big)\big) \\
&= \big(1 - \mathbb{E}[A_{p\,ij}]\big) \cdot \sigma(\ell_i + \ell_j) \cdot \big(1 - \sigma(\ell_i + \ell_j)\big) \\
&\quad + \delta_{ij} \sum_{k \in [n]} \big(1 - \mathbb{E}[A_{p\,ik}]\big) \cdot \sigma(\ell_i + \ell_k) \cdot \big(1 - \sigma(\ell_i + \ell_k)\big) \\
&= \big(1 - \mathbb{E}[A_{u\,ij}]\big) \cdot \sigma(\ell_i + \ell_j) + \delta_{ij} \sum_{k \in [n]} \big(1 - \mathbb{E}[A_{u\,ik}]\big) \cdot \sigma(\ell_i + \ell_k) \\
&= \mathbb{E}[A_{r\,ij}] \cdot \big(1 - \mathbb{E}[A_{u\,ij}]\big) + \delta_{ij} \sum_{k \in [n]} \mathbb{E}[A_{r\,ij}] \cdot \big(1 - \mathbb{E}[A_{u\,ik}]\big).
\end{aligned}
$$

The expression for the gradient in this context closely resembles the one presented in Chanpuriya et al. (2021), with the only variation being the substitution of $\mathbb{E}[A_u]$ matrices with additional $\mathbb{E}[A_r]$

matrices. This substitution is reasonable since the referenced work does not involve a primary sampled graph, and therefore the union graph corresponds precisely to the residual graph. This modified gradient translates directly into a modified fitting algorithm, pseudocode for which is given in Algorithm 2.

---

**Algorithm 2:** Fitting the modified odds-product model

**input** : target degrees $d \in \mathbb{R}^n$, primary expected adjacency $\mathbb{E}[A_p] \in [0,1]^{n \times n}$, error threshold $\epsilon$
**output** : symmetric probability matrix $\mathbb{E}[A_r] \in [0,1]^{n \times n}$

1   $\ell \leftarrow \mathbf{0}$             $\triangleright \ell \in \mathbb{R}^n$ is the vector of logits, initialized to all zeros
2   $\mathbb{E}[A_r] \leftarrow \sigma\left(\ell \mathbf{1}^\top + \mathbf{1}\ell^\top\right)$   $\triangleright \sigma$ is an entrywise logistic function, $\mathbf{1}$ is a length-$n$ all-ones column vector
3   $\mathbb{E}[A_u] \leftarrow \mathbf{11}^\top - \left(\mathbf{11}^\top - \mathbb{E}[A_p]\right) \circ \left(\mathbf{11}^\top - \mathbb{E}[A_r]\right)$         $\triangleright \circ$ is an entrywise product
4   $\hat{d} \leftarrow \mathbb{E}[A_u]\mathbf{1}$                $\triangleright$ expected degree sequence of $\mathbb{E}[A_u]$
5   **while** $\|\hat{d} - d\|_2 > \epsilon$ **do**
6      $B \leftarrow \mathbb{E}[A_r] \circ \left(\mathbf{11}^\top - \mathbb{E}[A_u]\right)$
7      $J \leftarrow B + \text{diag}\left(B\mathbf{1}\right)$       $\triangleright$ diag is the diagonal matrix with the input vector along its diagonal
8      $\ell \leftarrow \ell - J^{-1}\left(\hat{d} - d\right)$         $\triangleright$ rather than inverting $J$, we solve this linear system
9      $\mathbb{E}[A_r] \leftarrow \sigma\left(\ell \mathbf{1}^\top + \mathbf{1}\ell^\top\right)$
10     $\mathbb{E}[A_u] \leftarrow \mathbf{11}^\top - \left(\mathbf{11}^\top - \mathbb{E}[A_p]\right) \circ \left(\mathbf{11}^\top - \mathbb{E}[A_r]\right)$
11     $\hat{d} \leftarrow \mathbb{E}[A_u]\mathbf{1}$
12 **return** $\mathbb{E}[A_r]$

---

## C    FURTHER DETAILS ON EXPERIMENTAL SETTING

We now expand on the discussion of our experimental setting from Section 5.

**Statistics.** We choose 8 statistics that capture both the connectivity of a graph, as well local patterns within it. We look at the following network statistics:

- Pearson correlation coefficient (PCC) of the input and generated degree sequences (number of nodes incident to each node), as well as max degree.
- PCC of the input and generated triangle sequences (number of triangles each node belongs to).
- Normalized triangle, 4-clique, and 4-cycle counts.
- Characteristic (average) path length and fraction of node pairs which are connected. Letting $|C_i|$ be the size of $i$-th connected component, the latter quantity is $\sum_i \binom{|C_i|}{2} / \binom{n}{2}$).

**Datasets.** In our experiments we use the following eight publicly available datasets (together with one synthetic dataset) that we describe next. Table 2 also provides a summary of them.

- CITESEER: A graph of papers from six scientific categories and the citations among them
- CORA: A collection of scientific publications and the citations among them.
- PPI: A subgraph of the PPI network for Homo Sapiens. Vertices represent proteins and edges represent interactions.
- POLBLOGS: A collection of political blogs and the links between them.
- WEB-EDU: A web-graph from educational institutions.

We also include the following smaller sized datasets where we additional evaluate the GraphVAE method, as it is designed for datasets of that scale.

- LES MISERABLES: Co-appearance network of characters in the novel Les Miserables.
- FACEBOOK-EGO: A small-sized ego-network of Facebook users

- WIKI-ELECT: A (temporal) network of Wikipedia users voting in administrator elections for or against other users to be promoted to administrator. We use the first 15% of edges.

- RING OF CLIQUES: A union of 10 cliques of size 10 connected in a ring.

Table 2: Dataset summaries

| Dataset | Nodes | Edges | Triangles |
|---|---|---|---|
| CITESEER (Sen et al., 2008) | 2,110 | 3,668 | 1,083 |
| CORA (Sen et al., 2008) | 2,485 | 5,069 | 1,558 |
| PPI (Stark et al., 2010) | 3,852 | 37,841 | 91,461 |
| POLBLOGS (Adamic & Glance, 2005) | 1,222 | 16,714 | 101,043 |
| WEB-EDU (Gleich et al., 2004) | 3,031 | 6,474 | 10,058 |
| LES MISERABLES (Knuth, 1993) | 77 | 254 | 467 |
| WIKI-ELECT (Leskovec et al., 2010b) | 367 | 1,475 | 2,249 |
| FACEBOOK-EGO (Leskovec & Mcauley, 2012) | 324 | 2,514 | 10,740 |
| RING OF CLIQUES (synthetic dataset) | 100 | 4,600 | 1,200 |

**Hyperparameters and ranging overlap.** We range overlap in the following way for each method:

- MCEI, MCNI, MCFD: We range, respectively, the probability $p$ of including an edge of a clique, or node of a clique, or a clique in the sampled graph. We typically use evenly spaced numbers for $p$ over the interval between $0$ and $1$. For graphs with larger number of maximal cliques (PolBlogs and PPI), we additionally take the square of these values to decrease the probability.

- CELL method: We set the dimension of the embeddings at 32 for the larger datasets and 8 for the smaller ones. In order to range overlap, we follow the same early-stopping approach as in the original paper. Namely, we stop training at every iteration and sample 10 graphs every time the overlap between the generated graphs and the input graph exceeds some value (we set 10 equally spaced thresholds between 0.05 and 0.75).

- GraphVAE: We use 512 dimensions for the graph-level embedding that precedes the MLP and 32 dimensions for the hidden layers of the graph autoencoder. Again, we follow the early-stopping approach during training. We use the following implementation: `https://github.com/kiarashza/graphvae-mm`.

- VGAE: As this method relies on a dot-product kernel that has been shown to be unable to generate sparse and triangle-dense graphs, instead of fixing the embedding dimensions at a small number, we increase the dimensions of the hidden layers from 4 to 1024 and train for $5,000$ epochs. We use the publicly available implementation by the authors of this method: `https://github.com/tkipf/gae`.

**Implementation.** Our implementation of the methods we introduce is written in Python and uses the NumPy (Harris et al., 2020) and SciPy (Virtanen et al., 2020) packages. Additionally, to calculate the various graph metrics, we use the following packages: MACE (MAximal Clique Enumerator) (Takeaki, 2012) for maximal clique enumeration (it takes $O(|V||E|)$ time for each maximal clique) and Pivoter (Jain & Seshadhri, 2020) for $k$-clique counting.

## D    ADDITIONAL STATISTICS PLOTS

We also include plots for the statistics of the remaining networks: CITESEER, CORA, POLBLOGS, WEB-EDU, WIKIELECT, FACEBOOK-EGO, and the synthetic RING OF CLIQUES graph. The results largely follow the same pattern as results for the dataset presented in Section 5. Note that our baseline methods typically outperform other methods in generating a graphs with a higher number of triangles in a low overlap regime. This is especially more evident in sparse networks, e.g., compare CORA with POLBLOGS.

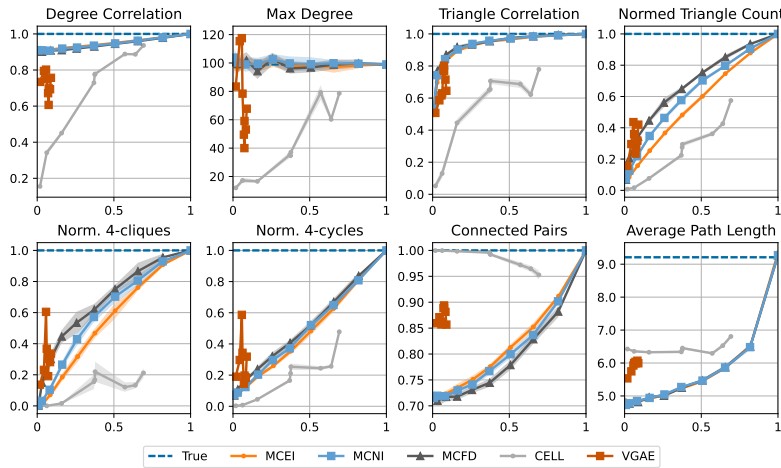

Figure 3: Statistics for CITESEER as a function of the *overlap*.

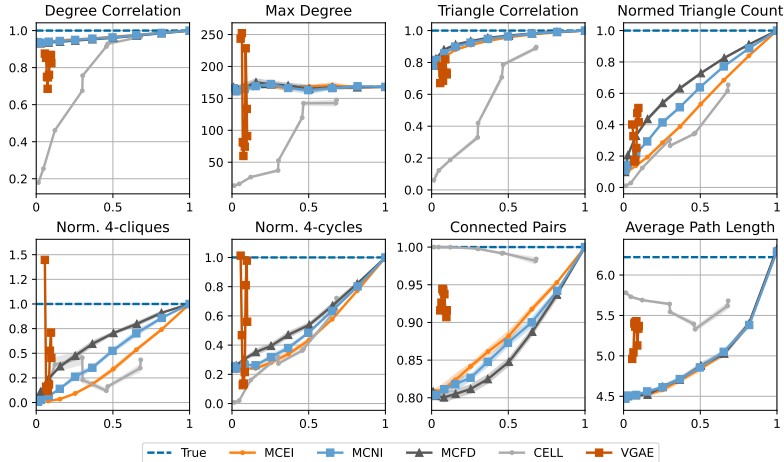

Figure 4: Statistics for CORA as a function of the *overlap*.

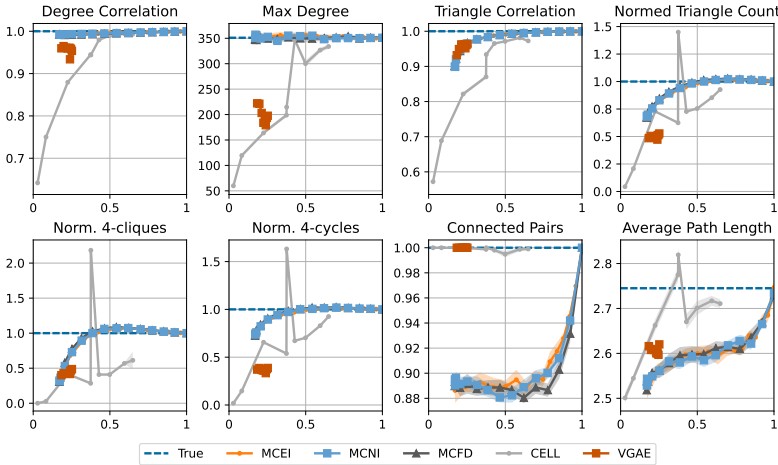

Figure 5: Statistics for POLBLOGS as a function of the *overlap*.

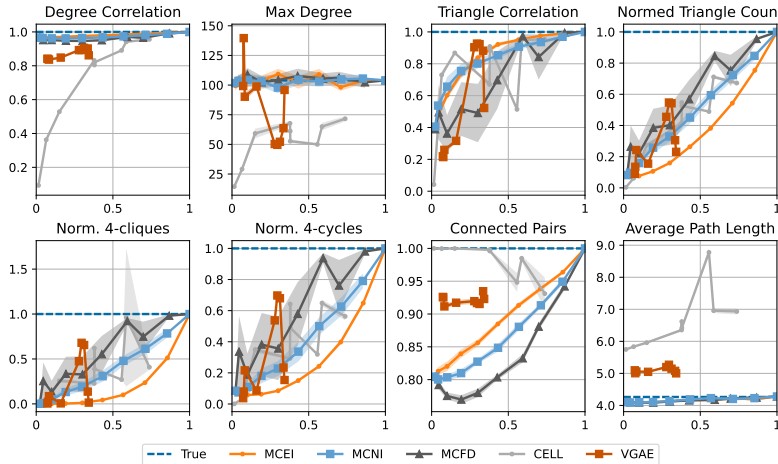

Figure 6: Statistics for WEB-EDU as a function of the *overlap*.

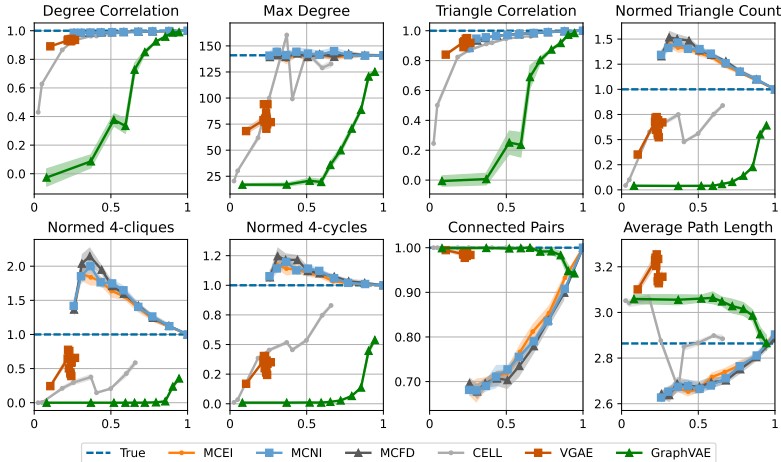

Figure 7: Statistics for WIKIELECT as a function of the *overlap*.

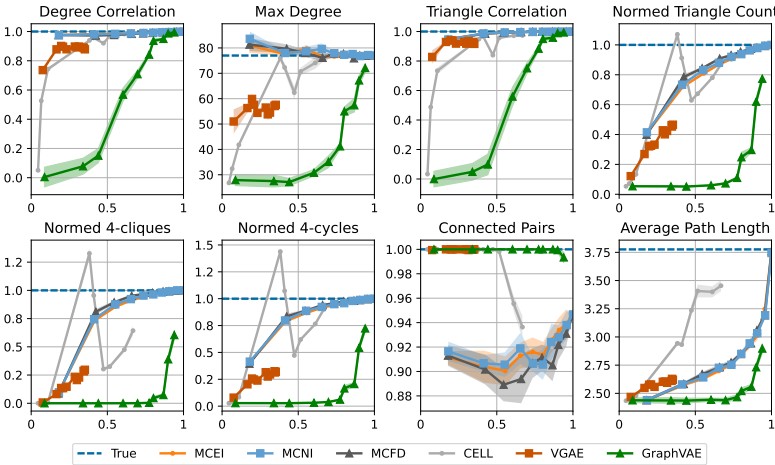

Figure 8: Statistics for FACEBOOK-EGO as a function of the *overlap*.

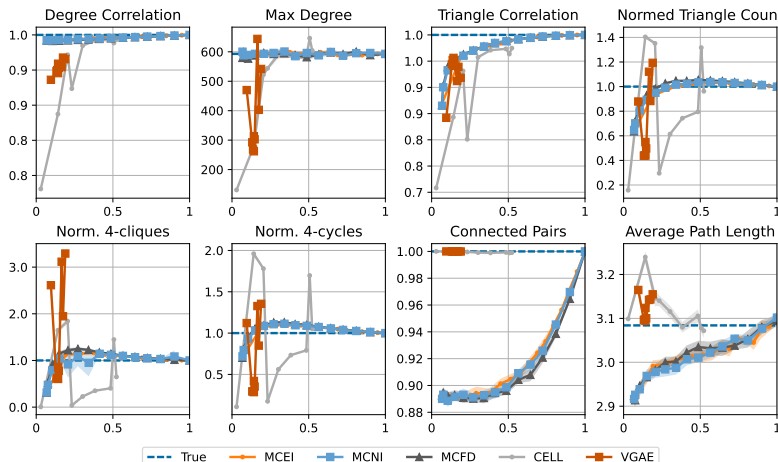

Figure 9: Statistics for PPI as a function of the *overlap*.

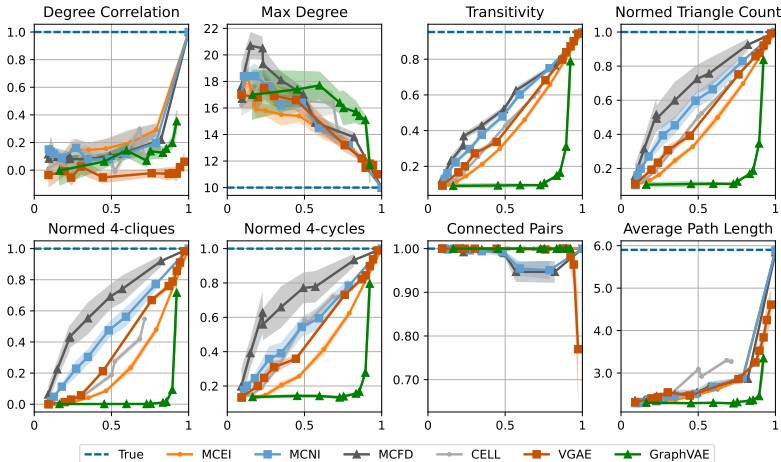

Figure 10: Statistics for the RING OF CLIQUES graph. All nodes are incident to the same number of triangles, so correlation is not defined for this graph. We replace it with transitivity.

# E    EMPIRICAL VALIDATION OF TIGHT FAMILY TRIANGLE BOUNDS

In Figure 11, we plot triangle counts for some instances of the tight random graph models described in Section 3. We observe that the tight examples achieve the theoretical upper bounds we prove in Section 4.

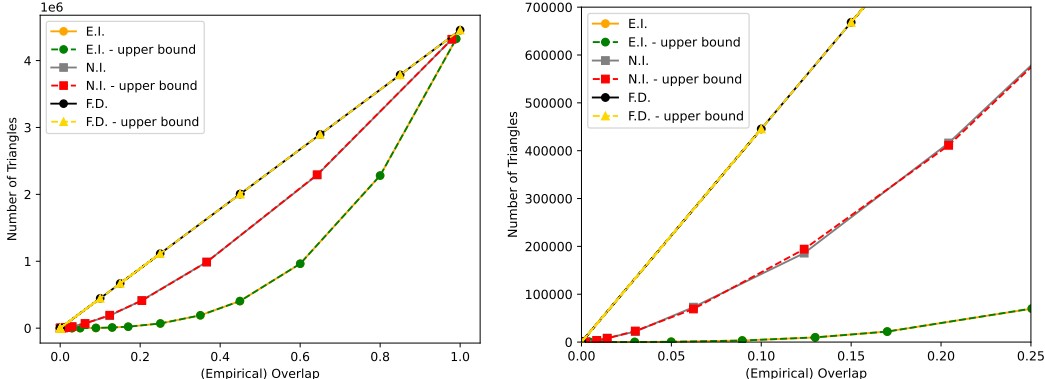

Figure 11: Triangle counts of "tight" random graph models and theoretical upper bounds. The plot on the right zooms in on the low overlap regime. Note that the triangle counts match the bounds.

