# OpenReview forum: "On the Role of Edge Dependency in Graph Generative Models"
_ICLR.cc/2024/Conference — Submitted to ICLR 2024_

### Official Review · Reviewer_ofBG · 2023-10-21

**Soundness:** 3 good
**Presentation:** 3 good
**Contribution:** 3 good
**Rating:** 6
**Confidence:** 4

**Summary:**

The paper categorizes generative models into three types of complexity: edge independent, node independent, and fully dependent. For each type and given the overlap of the graph model, the paper analyzes the upper bound in the number of triangles, obtaining a relation among them (EI, NI, and FD). The paper also proposes new generative models for each type of model based on a common algorithm.

**Strengths:**

The paper is original and of high quality. It proposes theoretical bounds considering the overlap of edges in generative models. The demonstrations of the theorems seem fine. The paper is also well-written and clear, especially the introduction. It is also significant regarding this specific topic.

**Weaknesses:**

The first phrase of the paper is misleading. It says: "we introduce a novel evaluation framework for generative models of graphs", but there is no evaluation framework.

Regarding the theoretical proposition, I do not have major concerns, but the generative models and results can be improved.

It is not clear the relation between the generative model with the theoretical contribution. The paper mentions: "We shift our focus towards empirically evaluating the real-world trade-off between overlap and performance across several specific models on real-world networks.". So, it seems a completely different paper.

The proposed models are trivial, so it does not seem to be an important contribution.  There is no training algorithm for p, and the search of the hyperparameter is a grid search.

As can be observed in the results, the models can not replicate the characteristics with low overlap. So, it does not seem to fulfill the second characteristic mentioned in the paper "A should exhibit low overlap". This is even worse with the other networks.

The details of the experiments are in the appendix, rather than the main paper.

**Questions:**

I suggest separating the two contributions. Personally, while I enjoy the first part of the paper, the generative model seems to be something that could lead to the rejection of the paper.

---

> ### Author Response · Authors · 2023-11-17
> **Response to Reviewer ofBG**
>
> We would like to thank you for your detailed comments and for appreciating the novelty of our theoretical bounds. We address all comments below:
>
> > The first phrase of the paper is misleading. It says: "we introduce a novel evaluation framework for generative models of graphs", but there is no evaluation framework.
>
> The evaluation framework that we refer to is that of evaluating graph generative models under the lens of overlap. We believe this angle is missing from the evaluation of graph generative models, in general. Typically, models are evaluated in their ability to mimic properties of the input graph, without the important constraint of generating diverse (low-overlap) graphs. This limits the effectiveness of evaluation, since by simply memorizing the input graph, a model can perfectly mimic its properties, but not be a useful generative model.
> For example, NetGAN/CELL is known to be able to simply reproduce the input graph. Hence, the authors suggest an early stopping criterion to prevent this from happening. This suggests that early stopping/not reproducing the input graph was an important criteria for their model design. However, this criteria was not formalized or measured in the evaluation of it. Our work formalizes this criteria and advocates using it in graph generative model evaluation going forward.
>
> > Regarding the theoretical proposition, I do not have major concerns, but the generative models and results can be improved.
>
> Indeed our theoretical results are tight as we demonstrate with examples and can not be improved further. Regarding the generative models, our target is not to provide a state-of-the-art model, but contribute in two other significant ways: (i) in addition to providing an evaluation framework for studying graph generate models under the overlap criteria, we also provide models for which it is easy to control the overlap and do not rely on heuristic methods, such as early stopping, and (ii) show that even those simple models can perform comparably well to more complex (DL) models in matching graph statistics under a low overlap regime.
>
> > It is not clear the relation between the generative model with the theoretical contribution. The paper mentions: "We shift our focus towards empirically evaluating the real-world trade-off between overlap and performance across several specific models on real-world networks.". So, it seems a completely different paper.
>
> Our paper consists of two contributions that we believe are connected: (i) provides theoretical bounds on the power of different graph generative models (under the hierarchy we propose) with respect to the overlap criterion, and (ii) an evaluation of the ability of different models (including our own baselines) to mimic properties of the input graph. As the models we choose fall under different levels of the hierarchy (same for the models we construct) and the evaluation is done under the overlap criterion, we believe these two contributions are strongly connected.
>
> > The proposed models are trivial, so it does not seem to be an important contribution. There is no training algorithm for p, and the search of the hyperparameter is a grid search.
>
> The proposed models are simple as our target is not to provide a state-of-the-art model, rather to show that even simple models can perform comparably to complex DL-based models, when we restrict the latter not to simply memorize the input graph. Regarding p, there is no need for grid search. “p” simply controls the amount of overlap between the input and the generated graphs. Naturally, the higher the overlap (thus the value of “p”), the better the generated graphs will mimic the input graph. Hence, “p” is not a value to be fine-tuned, rather it is left to a practitioner to decide how much overlap will allow.
>
> > As can be observed in the results, the models can not replicate the characteristics with low overlap. So, it does not seem to fulfill the second characteristic mentioned in the paper "A should exhibit low overlap". This is even worse with the other networks.
>
> The task of replicating characteristics under a low overlap is a hard one as our theoretical results demonstrate. Notice how even deep learning models fail at low overlap. Under our evaluation framework, the overall performance of these deep models is not clearly better than simpler models which are more interpretable and easier to control the overlap.
>
> >The details of the experiments are in the appendix, rather than the main paper.
>
> Evaluating all of these methods in the common framework that we propose involves a lot of details, many of which are not crucial to the core takeaways of the work. We believe it is fairly standard to defer such details to the appendix, and given that the supplementary material will be published jointly with the paper, it ensures replicability of our results.

---

### Official Review · Reviewer_V9nt · 2023-10-30

**Soundness:** 3 good
**Presentation:** 3 good
**Contribution:** 3 good
**Rating:** 8
**Confidence:** 4

**Summary:**

This paper proposes a new framework for generative models for graphs. Given an input graph $G_i$, this framework provides three model types such that, for a fixed planting probability $p$ and a fixed model type, it outputs a random graph $G_p$ with the degree sequence that is expected to be the same with $G_i$. $G_p$ is expected to have as many as similar statistics to $G_i$, e.g., normed number of small cliques and small cycles, while keeping the edge diversity that is measured by the overlap of this random graph model. The model types include edge independent (EI), node independent (NI) and fully dependent (FD) models, which is a hierarchical categorization of random graph generative models. Theorems 1-3 provide upper bounds of the expected number of triangles in terms of overlap parameter for these three models, respectively. The experimental results on several real datasets and a synthetic dataset demonstrate the superiority of this method.

**Strengths:**

(1) The three hierarchies of graph generative model are interesting and worth further study.

(2) The theoretical results for the upper bounds and the tight examples of expected triangle counts for three model types are novel.

**Weaknesses:**

(1) Overlap seems not to be a good meausre for diversity of small graphs since it does not take isomorphism of graphs into account.

(2) The effectiveness of this method on large graphs is unknown.

(3) The practical use of these models such as those introduced in the first paragraph of this paper is not clear.

**Questions:**

(1) How do you select maximal clique one by one in Algorithm 1? Is there overlap between two max cliques?

(2) If the input graph is sparse enough such that the connectivity is weak (for example, most node degrees are 1 or 2), then intuitively, the sampled graph $G_p$ will be broken up seriously. Even after adding a sampled second graph $G_r$, how does Algorithm 1 guarantee the statistics of $G_u$?

(3) Why the statistics results of GraphVAE are missing in Figures 3-6 and 9?

(4) How do you generate the small subgraphs of benchmarks in Table 2? Is there a unified approach for this?

(5) How large a graph can this method apply to? Do the experiments not use large datasets just because it is difficult to verify the results?

(6) Is there manifest applications of your framework in practical use, e.g., spread in social for financial networks or drug discovery?

---

> ### Author Response · Authors · 2023-11-17
> **Response to Reviewer V9nt**
>
> Thank you for your thoughtful review and for finding the proposed hierarchy of graph generative models interesting. We address all comments and questions below:
>
> > Overlap seems not to be a good measure for diversity of small graphs since it does not take isomorphism of graphs into account.
>
> Thanks – this is a great point. We have considered the same issue before – one can avoid it by defining a notion of overlap that looks at the maximum edge overlap between two graphs taken over any permutation of the node ids. Unfortunately, this is a very hard metric to compute in both theory and practice, due to its connection to the graph isomorphism problem. Developing a useful metric inspired by this approach that can be computed efficiently, and studying its theoretical properties, would be a very nice direction for future work.
>
> > (1) How do you select maximal clique one by one in Algorithm 1? Is there overlap between two max cliques?
>
> We use a fast state of the art implementation for maximal clique listing, more specifically [MACE](https://research.nii.ac.jp/~uno/code/mace.html). Indeed, two maximal cliques can overlap. E.g., two 4-cliques can overlap on the 3 common nodes, but each one has a separate node not connected to each other. Hence, edges/nodes common among 2 (or more) maximal cliques have higher probability of being sampled, helping the benchmarks achieve high triangle counts in low overlap.
>
> > (2) If the input graph is sparse enough such that the connectivity is weak (for example, most node degrees are 1 or 2), then intuitively, the sampled graph G_p will be broken up seriously. Even after adding a sampled second graph G_r, how does Algorithm 1 guarantee the statistics of  G_u?
>
> Nodes of degree 1, or 2, can still be part of a maximal clique (even an edge is a maximal clique if no node can be added to it to make it a clique). Hence, there is a positive probability that edges incident to those nodes will be sampled. Indeed, however, those nodes may end up disconnected. It would be interesting to investigate strategies that can increase the connectivity of the graph while maintaining the degree distribution.
>
> > (3) Why the statistics results of GraphVAE are missing in Figures 3-6 and 9?
>
> GraphVAE is a method designed for generating small networks (GraphVAE: Towards Generation of Small Graphs Using Variational Autoencoders), hence it faces scalability issues for moderate to large sized graphs. For this reason, we only evaluate this method on a subset of small-sized datasets. Notice that this method employs a fully connected MLP to generate all $n \choose 2$ possible edges, hence for graphs >1K we face both runtime and memory issues.
>
> > (4) How do you generate the small subgraphs of benchmarks in Table 2? Is there a unified approach for this?
>
> Yes, we use a unified approach for all datasets. The approach is based on using a maximal clique enumeration algorithm (see answer to 1), followed by Algorithm 1 that samples either (i) each edge of a maximal clique independently for the EI benchmark, (ii) each node of a maximal clique independently for the NI benchmark, or (iii) each maximal clique independently for the FD benchmark.
>
> > (5) How large a graph can this method apply to? Do the experiments not use large datasets just because it is difficult to verify the results?
>
> The scalability of this approach depends on the maximal clique listing algorithm used. We could easily scale to graphs of ~4-5k nodes, which is comparable (or larger) to the size of graphs that DL methods can scale to (e.g., NetGAN/CELL, or GraphRNN). For applications to larger networks, we note that our models only need some groupings of the input graph’s edges, not necessarily its maximal cliques. Therefore, edge groupings like dense subgraph discovery, decomposition methods (Elfarouk et al. 2022), and near-clique discovery (Tsourakakis et al. 2013), which are tractable to compute, could be used instead of max cliques.
>
> >(6) Is there manifest applications of your framework in practical use, e.g., spread in social for financial networks or drug discovery?
>
> Our methods and theoretical results highlight powers and limitations of three classes of graph generative models. We provide theoretical results for the ability (or shortcomings) of each class of models to generate diverse graphs that are triangle (or k-cycle) dense. We believe our findings are useful in applications that rely on generating diverse graphs under certain constraints.
>
> Drug discovery is such an example of constrained  graph generation, where diversity is also crucial. Our results are also especially relevant to modeling social networks, which typically exhibit high triangle density. Generating realistic and diverse social network graphs is important when privacy concerns arise, when testing the robustness of community detection algorithms, and beyond.

---

### Official Review · Reviewer_4Knv · 2023-10-31

**Soundness:** 3 good
**Presentation:** 3 good
**Contribution:** 2 fair
**Rating:** 5
**Confidence:** 4

**Summary:**

This paper presents an evaluation framework for graph generative models leveraging the graph overlap, which ensures both accuracy and edge-diversity. Also, they provide proofs the bounds of number of triangles and other short-length cycles based on the model overlap. Finally, they categorize graph generative models into three categories: : edge independent, node independent, and fully dependent models and provide new generative models for each category.

**Strengths:**

1. Proved the triangle count bounds for each generative model category leveraging overlaps.
2. Presented simple graph generative models without using complex deep generative model architecture.

**Weaknesses:**

1. Three categorizations for graph generative models cannot include the most recent generative models. Can you give more examples of recent generative models (e.g., GDSS, GraphARM, DiGress, etc.) that fit the proposed categorization?
2. The definitions for EI, FD, NI models are vague, and a more formal definition for categorization is needed. For instance, is the definition of FD model formal? It’s hard to understand what “the generative models allow for any possible distribution A” means.
3. A detailed explanation of the strength of using overlaps for graph generative models is missing. The authors emphasize that accuracy and diversity are key characteristics to evaluate the graph generative models but it’s hard to understand why overlap can be a good evaluation framework.
4. Lack of comparison to commonly used metrics such as MMD or V.U.N. MMD and V.U.N is the most popular evaluation metric for graph generative models. Still, I cannot find any comparison of overlap to current evaluation metrics for graph generative models.

**Questions:**

1. Why do we need bounds or theoretical limits on the number of triangles and other short-length cycles? I cannot understand the reason for the existence of the boundary and their proofs.
2. Hard to understand the key contribution of this paper. Is it right that the proposed evaluation framework is to compare the overlap between generated graphs and test graphs?
3. Is triangle-based evaluation limited to the models that contain many triangles? For instance, for grid graph generation, can the proposed method capture the graph generation quality well?
4. What does the x-axis mean in Figure 2? I can briefly understand that the lines following the dotted line (True) are desirable but cannot understand what Figure 2 means exactly.

---

> ### Author Response · Authors · 2023-11-16
> **Response to Reviewer 4Knv (1/2)**
>
> Thank you for the thoughtful review. We address the comments and critiques below.
>
> >Can you give more examples of recent generative models (e.g., GDSS, GraphARM, DiGress, etc.) that fit the proposed categorization?
>
> Similarly to ERGMs, auto-regressive approaches (like GraphARM, and including diffusion models like DiGress) fall into the FD category of our hierarchy. Depending on the mapping function, flow-based models (like GraphAF) are likely to be FD as well.
>
> > The definitions for EI, FD, NI models are vague, and a more formal definition for categorization is needed. For instance, is the definition of FD model formal? It’s hard to understand what “the generative models allow for any possible distribution A” means.
>
> We believe our definitions of the EI and NI model classes are formal and unambiguous, though we are open to any specific criticisms. Regarding the FD model definition, indeed, it may be more clear to rephrase as “Any categorical distribution $\mathcal{A}$ over adjacency matrices $\boldsymbol{A} \in \{0,1\}^{n \times n}$ of undirected graphs without self-loops is an FD graph generative model.” We hope this addresses any concern about ambiguity. The FD definition will necessarily be sparser than the others, since NI and EI are carefully-constructed subsets of FD.
>
> > A detailed explanation of the strength of using overlaps for graph generative models is missing. The authors emphasize that accuracy and diversity are key characteristics to evaluate the graph generative models but it’s hard to understand why overlap can be a good evaluation framework.
>
> Overlap is perhaps the simplest characterization of the (lack of) diversity of graphs generated by a graph generative model - roughly, it is the fraction of edges shared between two sampled graphs, in expectation. A graph generative model having high overlap means that sampling from it tends to generate similar sets of edges, rather than a diverse pool of edges. Consequently, requiring low overlap is a simple way of preventing a graph generative model from achieving high accuracy (high fidelity to the input graph, e.g., in terms of a high triangle count) by simply outputting the same or a very similar set of edges as the input graph.
>
> > Lack of comparison to commonly used metrics such as MMD or V.U.N.
>
> These metrics are not appropriate for the setting we test, which involves generation of a graph that is similar to a single input graph. Note that we test the same setting considered in the Variational Graph Auto-Encoder (Kipf and Welling, 2016) and NetGAN (Bojchevski et al., 2018; Rendsburg et al., 2020) papers, among others. By contrast, MMD is suitable for comparison of distributions of graphs, that is, a setting with multiple input graphs and multiple out graphs. VUN is also intended for such a setting, specifically for molecule generation.

---

> ### Author Response · Authors · 2023-11-16
> **Response to Reviewer 4Knv (2/2)**
>
> >Why do we need bounds or theoretical limits on the number of triangles and other short-length cycles?
>
> Triangle density is a hallmark property of several types of networks. Closely related is the idea of triadic closure, that if a node A links to a node B and also to a node C, then B and C themselves are likely to be linked. This is a fundamental characteristic of social networks, among other kinds of networks, and plays an important role in network analysis. If a graph generative model is limited in its capacity to produce graphs with many triangles, it is limited in its capacity to generate many kinds of realistic networks. These bounds therefore help characterize the power of graph generative models. It has been an important goal of recent work in the area to understand theoretical limits on the ability of various graph models to represent triangle-dense graphs — for example, see the work of Seshadhri et al. (2020) on “The impossibility of low-rank representations for triangle-rich complex networks.”
>
> >Is it right that the proposed evaluation framework is to compare the overlap between generated graphs and test graphs? …What does the x-axis mean in Figure 2? I can briefly understand that the lines following the dotted line (True) are desirable but cannot understand what Figure 2 means exactly.
>
> The x-axis in Figure 2 is the overlap of a model that was fit to an input graph; each curve represents a different method with which to fit a model, such as GraphVAE. The y-axis represents accuracy, since, as you note, it shows how closely these models match some statistic, such as triangle count, of the input graph, which is shown by the “True” dotted line. Therefore, these are essentially several plots of accuracy vs diversity. What we find is that methods overall struggle to attain high accuracy while maintaining high diversity, and as the diversity requirement is decreased even simple methods like the ones we propose can be competitively accurate as some prior deep ones.
>
> > Is triangle-based evaluation limited to the models that contain many triangles? For instance, for grid graph generation, can the proposed method capture the graph generation quality well?
>
> The primary advantage of our proposed methods is that they are simple and allow for directly controlling overlap. They can capture any kind of graph well in the sense that if there is no restriction on overlap/diversity, by setting $p=1$, they simply output the input graph and hence capture all of its statistics. When required to generate a diverse set of graphs, however, our clique-based methods may not be suitable for certain kinds of graphs like grid graphs. Clique-based graph generation in general is more suitable for graphs like social networks, which have strong community structure.

---

> > ### Comment · Reviewer_4Knv · 2023-11-20
> >
> > I slightly revised the score as the authors kindly responded to my concerns and questions. However, I have some remaining concerns and questions.
> >
> > The related works for deep graph generative models including GraphGAN, GraphRNN, MolGAN, and GraphVAE make it harder to understand the setting with a single input graph, not multiple input graphs. Clearly explaining that the work is targeting the single graph input task and including the related works such as GraphAE may solve this issue.
> >
> > In addition, to the best knowledge, V.U.N is not intended for multiple input graphs. Specifically, validity and uniqueness are computed only with generated graphs regardless of input, and novelty is computed with respect to the training graphs by isomorphic test. I understand that the novelty is not that appropriate for a single input graph setting, at least the validity and uniqueness could be evaluated as the authors emphasize that accuracy and diversity are key characteristics to evaluate graph generative models. Of course, I understand that the validity could not be computed for any datasets, additional experiments for the large graphs that enable the computation of the validity (e.g., SBM data) may make the work more persuasive.

---

> > > ### Author Response · Authors · 2023-11-21
> > > **Response to Reviewer 4Knv Comment**
> > >
> > > We would like to thank you for reading our response and for providing additional insightful comments.
> > >
> > > >  The related works for deep graph generative models including GraphGAN, GraphRNN, MolGAN, and GraphVAE make it harder to understand the setting with a single input graph, not multiple input graphs. Clearly explaining that the work is targeting the single graph input task and including the related works such as GraphAE may solve this issue.
> > >
> > > Thank you for this suggestion. We will follow your advice and revise the introduction and related works sections to make it clear that our work targets the single input graph task.
> > >
> > > > In addition, to the best knowledge, V.U.N is not intended for multiple input graphs. Specifically, validity and uniqueness are computed only with generated graphs regardless of input, and novelty is computed with respect to the training graphs by isomorphic test. I understand that the novelty is not that appropriate for a single input graph setting, at least the validity and uniqueness could be evaluated as the authors emphasize that accuracy and diversity are key characteristics to evaluate graph generative models. Of course, I understand that the validity could not be computed for any datasets, additional experiments for the large graphs that enable the computation of the validity (e.g., SBM data) may make the work more persuasive.
> > >
> > > Thank you for your points! Indeed, the VUN measure, similarly to overlap, encourages diversity through uniqueness and can also be applied to the single input graph case. We would like to thank you for bringing this to our attention. We will add a discussion related to this measure and overlap. We notice two key differences among overlap and uniqueness: 1) “Uniqueness” counts the fraction of generated graphs belonging to a unique isomorphism class. Hence, it looks at exactly overlapping graphs (after isomorphism) and doesn’t, e.g., prevent the model from always returning graphs that are essentially the same with a few perturbations. 2) In contrast to overlap, it is a very expensive measure to compute on large graphs as it requires solving the graph isomorphism problem.
> > >
> > > We can actually extend the overlap measure to account for graph isomorphisms by defining a notion of it that looks at the maximum edge overlap between two graphs taken over any permutation of the node ids. This prevents trivial solutions of almost isomorphic graphs that Uniqueness can not handle. Again, though, this is a very hard measure to compute in both theory and practice. Developing a useful measure inspired by this approach that can be computed efficiently, and studying its theoretical properties, would be a very nice direction for future work.
> > >
> > > As you also point out, validity requires knowledge of some ground-truth class. For the datasets we consider it is almost impossible to define some class. Instead, we look at certain properties these networks have. For example, social networks are known to be rich in triangles, follow a specific degree distribution (power-law) and have small diameter. Our statistics in Figures 2 and afterwards look at those specific properties and how close the generated networks are to those properties. Hence, this is a proxy to measuring the “Validity” of the generated graphs into following certain properties the input graphs have.

---

### Official Review · Reviewer_AsDh · 2023-10-31

**Soundness:** 3 good
**Presentation:** 3 good
**Contribution:** 3 good
**Rating:** 6
**Confidence:** 4

**Summary:**

The paper discusses the introduction of a novel evaluation framework for generative models of graphs centered around model-generated graph overlap to capture both accuracy and edge diversity. It is based on the theoretical analysis of a hierarchy of graph generative models that analyze the theoretical bounds on short-length cycles (though the paper focuses on triangle counts in the theoretical analyis and other cycles in the experiments) production based on model overlap. This analysis is done concerning three types of models: node-independent, fully independent, and node-independent models. The paper also introduces new generative models. The experiments measure the quality and overlap of the proposed models and a comparative evaluation against six models over 8 characteristics is performed using 8 real networks and 1 synthetic scenario. The proposed models show competitive performance.

**Strengths:**

The paper is interesting from at least three perspectives. First, it is good to have a formalization for independence, independence that is not reliant on the mechanics of the generative model and thus could be applicable in a broader sense. However, this is not totally new as there are existing approaches such as exchangeability (node and edge) that provide insights concerning sampling order. Another interesting aspect of the paper is the idea of linking (in the theorems) the expected number of triangles with the overlap of two sampled graphs (expected number of shared edges). This could be very useful to provide greater control in clique generation on synthetic graphs (which is what is evaluated in the paper as well) among other benefits. A third interesting contribution is the proposed models. Although more clarity could be beneficial to the generative process. More specifically, the process of association probabilities to the locations within the graph (either adjacency matrix, locations, or edge lists) needs to be spelled out and better described.

**Weaknesses:**

There are a few things that could improve the paper. First, I am concerned about the cubic nature of the bounds. These seem to be very loose and it is hard to see how these can be used in practice, or even to provide insight about how to structure the generative process. Given this subtlety, something that could facilitate seeing the applicability of the bounds would be to see some synthetic verifications but no experiments show the application of the bounds to verify their veracity of them. This issue is especially important for the case of edge-independent models (cubic in the product $n \cdot Ov(\mathcal A$)). Other things could be done to improve the paper. For instance, I think it is important to highlight the methodological complexity of using the bounds to achieve the target structure in the sampled graphs. Currently, Algorithm 1 has one line that describes MCDF and a couple that describes MCEI and MCNI while most of details are not fully detailed in the main body of the paper.  That is, as I mentioned before, the process of association probabilities to the locations within the graph could be further clarified. Finally, identifying cycles can be computationally expensive and a scalability analysis of the method could be useful.

**Questions:**

In addition to the questions I wrote in the weaknesses section, I have the following doubts:

What is the cost of a growing number of nodes to the technique proposed for bounding the number of triangles?

What is the cost of the proposed models as the number of nodes grows?

How do you link the bounds to determine what edges have to be sampled to achieve a target structure?

---

> ### Author Response · Authors · 2023-11-16
> **Response to Reviewer AsDh (1/2)**
>
> Thank you for the thoughtful and positive review. We appreciate the recognition of our theoretical framework, triangle bounds, and proposed methods. We address your concerns below.
>
> > First, I am concerned about the cubic nature of the bounds. These seem to be very loose and it is hard to see how these can be used in practice, or even to provide insight about how to structure the generative process… What is the cost of a growing number of nodes to the technique proposed for bounding the number of triangles?
>
> Indeed, the bounds for the triangle count in terms of node count and overlap are cubic in the node count. This is not entirely surprising: the total number of possible triangles is also cubic in the node count (specifically, it is $\binom{n}{3}$), and we are showing that a model’s overlap bounds the fraction of these triangles that can appear in expectation. Further, we provide examples demonstrating the tightness of our triangle count bounds. These examples show that there must necessarily be a term that is cubic in the node count, at least in the absence of further constraints. Under specific models, we may expect overlap to scale inversely with the number of nodes: for example, with Erdős–Rényi random graphs or related models such as stochastic block model graphs, assuming we fix the average number of friends (i.e., expected node degree), the overlap scales as $O(n^{-1})$. Thus, our bounds scale subcubicly in $n$.
>
> Importantly, we note that the triangle bounds in terms of overlap and node count are not the tightest bounds that we provide. We also provide tighter bounds in terms of overlap and volume (i.e., expected number of edges) for EI and NI models - see the proof of Theorem 1 on page 6, prior to the application of Lemma 1. These overlap/volume bounds can be significantly tighter, but we apply Lemma 1 and convert them to the simpler overlap/node count bounds to highlight the conceptual differences between different levels of our proposed hierarchy, which is our focus.
>
> > something that could facilitate seeing the applicability of the bounds would be to see some synthetic verifications but no experiments show the application of the bounds to verify their veracity of them.
>
> Thank you for the useful suggestion. As you point out, a simulation can corroborate the tightness of our bounds. In Appendix E, we have added a plot with a simulation that shows that the example models indeed achieve the tight bounds.
>
> > I think it is important to highlight the methodological complexity of using the bounds to achieve the target structure in the sampled graphs… the process of association probabilities to the locations within the graph (either adjacency matrix, locations, or edge lists) needs to be spelled out and better described.
>
> The sampling of a graph with any of our proposed methods (MCEI, MCNI, and MCFD) involves sampling a primary graph $G_p$ and a residual graph $G_r$. Essentially all of the novel content of our methods — specifically 1) the use of max cliques, 2) the addition of edge dependency in the MCNI and MCFD models, and 3) the ability to easily control overlap — occurs in the sampling of $G_p$. We are able to compactly describe exactly how $G_p$ is sampled in Algorithm 1 due to the simplicity of this procedure. However, due to space constraints, we unfortunately have to appendicize Algorithm 2, which explains the sampling of $G_r$. In brief, Algorithm 2 samples a $G_r$ from a simple distribution such that the union of $G_r$ and $G_p$ (i.e., the union of their edge sets) matches the node degrees of the input graph in expectation.
>
> To the extent that we can assign probabilities to entries of the sample’s adjacency matrix, we can say this: For any given pair of nodes $i$ and $j$, if $p_p$ and $p_r$ are the probabilities that the edge $(i,j)$ occurs in $G_p$ and $G_r$, respectively, then the probability that edge $(i,j)$ occurs in the final sample (the union of $G_p$ and $G_r$) is $1 - (1-p_p)(1-p_r)$. Note that this is just a marginal probability; in the case of MCNI and MCFD, there is dependency between the appearance of different edges.

---

> > ### Author Response · Authors · 2023-11-16
> > **Response to Reviewer AsDh (2/2)**
> >
> > > What is the cost of the proposed models as the number of nodes grows? …a scalability analysis of the method could be useful.
> >
> > Our proposed generative models are based on maximal clique enumeration, which determines their scalability. This is theoretically intractable in the worst case, but often practical for small-to-mid-sized real-world networks, such as the ones in this paper. For applications to larger networks, we note that our models only need some (possibly overlapping) groupings of the input graph’s edges, not necessarily its maximal cliques. Therefore, edge groupings like dense subgraph discovery, decomposition methods (Elfarouk et al., 2022), and near-clique discovery (Tsourakakis et al., 2013), which are tractable to compute, could be used instead of max cliques.
> >
> > >How do you link the bounds to determine what edges have to be sampled to achieve a target structure?
> >
> > The process is quite straightforward given our bounds: the upper bound for the triangle count in terms of volume and overlap of a model can be solved for a lower bound for the overlap in terms of the triangle count and volume. For a given input graph, we know the latter two target quantities, so we can solve for the former.

---

> > > ### Comment · Reviewer_AsDh · 2023-11-22
> > > **Thank you for your reply**
> > >
> > > Thank you for your answers. I am sure that your clarifications about MCEI, MCNI, and MCFD, as well as the effect of the number of nodes will improve the paper. Please include your clarifications in the main body of your paper. There are two things that I think still need work. First how to use the theoretical bounds to determine what edges have to be sampled to achieve a target structure requires more clarity. A pseudocode could help with that. Second, the cubic bound still needs clarification. I will give a specific example of why your answer is still not clear on the importance of your contribution. Consider Erdos-Renyi graphs, which have binomial degree distributions (because of the Bernoulli parameterization of the model). The graphs sampled from such a model are hardly fully connected components -- which is known from the works of Tamara Kolda's lab and others (including Fan Chung and others), i.e., that the communities created with the ER model are highly disconnected and have no well-defined community structure. To create a community structure, additional constraints on the model are needed -- which have been widely proposed. Thus, it is quite obvious that a bound on the number of triangles (needed to produce a community structure) is cubic in the number because that is what you need to create cycles of length 3 -- three edges. So, the tighter bounds in terms of overlap and volume that you mention need further clarification to explain the novelty of your contribution.

---

> > > > ### Author Response · Authors · 2023-11-22
> > > >
> > > > Thank you for following up with further discussion. We are glad to look into integrating parts of the discussion into the revision to clarify our contribution. We are also glad to clarify parts of our paper and rebuttal below.
> > > >
> > > > > First how to use the theoretical bounds to determine what edges have to be sampled to achieve a target structure requires more clarity. A pseudocode could help with that.
> > > >
> > > > Our upper bounds on expected k-cycle count are impossibility results and do not directly correspond to any algorithm; they do not specify "what edges have to be sampled," but they do, roughly speaking, relate to "how many edges have to be sampled." (Apologies for any confusion from the prior response — we thought you indeed meant to ask the latter.) Perhaps relatedly to your comment, in Section 3, we do additionally show how to construct instances of 3 models — one in each of the EI, NI, and FD classes — that achieves these bounds, showing that they are tight.
> > > >
> > > > > Second, the cubic bound still needs clarification. I will give a specific example of why your answer is still not clear on the importance of your contribution. Consider Erdos-Renyi graphs... it is quite obvious that a bound on the number of triangles (needed to produce a community structure) is cubic in the number because that is what you need to create cycles of length 3.
> > > >
> > > > Indeed, it is well-known that for an Erdős–Rényi model, the expected triangle count is bounded above in $O(n^3)$ (and also, more tightly, in $O(\text{Vol}(\mathcal{A})^{3/2})$). Perhaps there is some misunderstanding about the scope of our results. Our results involve proving bounds on whole classes of models rather than specific kinds, and additionally showing that these bounds are made tighter when the model’s level of memorization, in terms of overlap, is bounded. The Erdős–Rényi model is a specific kind of edge independent model, and related prior work (“On the Power of Edge Independent Graph Models”) shows that the expected triangle count for *any* EI model is $O(n^3 \cdot \text{Ov}(\mathcal{A}))^3$. Note that since $\text{Ov}(\mathcal{A}) \in [0,1]$, this is a tighter bound than $O(n^3)$. Our theoretical contribution in this paper is to expand the notion of overlap to models beyond the EI class, including to models which allow for complex forms of dependency between edges, and prove similar bounds for these classes.

---

### Meta-Review · Area_Chair_CDVY · 2023-12-06

**Metareview:**

This article discusses a categorisation of generative models of graphs, providing upper bounds on the expected number of triangles with respect to a notion of model overlap. It also introduces some specific generative models corresponding to each category. The reviewers found the work interesting, but listed a number of limitations in the current submission. Reviewer 4Knv mentioned that some of the definitions were rather vague, with which I agree. As I understand from the author's response, the fully dependent category includes “Any categorical distribution over adjacency matrices of undirected graphs without self-loops is an FD graph generative model.” This means it actually encompasses all generative models of simple graphs, and  therefore includes the classical Erdos-Renyi, where there is no dependence. This section would need further clarification and polishing. Additionally, the relevance of the use of the overlap could also be better explained, as mentioned by Reviewer 4Knv. In my opinion, the author's response did not fully address this concern.

Finally, this article is missing key references on the literature on random graphs.
1/ The so-called "edge independent model", attributed to Chanpuriya et al. is known in the probability/statistics literature as the inhomogeneous Erdos-Renyi model, see e.g.
O. Klopp et al. Oracle inequalities for network models and sparse graphon models. Annals of Statistics, 2017.
F. Benaych-Georges et al. Largest eigenvalues of sparse inhomogeneous Erdos-Renyi graphs. Annals of Probability, 2019
C. Le et al. Concentration and regularization of random graphs. Random Structures and Algorithms, 2017.
2/ The so-called "node-independent model" is essentially a graphon model, also known as exchangeable random graph model. See e.g.
P. Orbanz and D. Roy. Bayesian Models of Graphs, Arrays and Other Exchangeable Random Structures. IEEE PAMI, 2013.
Diaconis, P. and Janson, S. (2008). Graph limits and exchangeable random graphs. Rendiconti di Matematica,
Serie VII, 28, 33–61.
L. Lovasz. Large Networks and Graph Limits. American Mathematical Society Colloquium Publications 60. Amer. Math. Soc., Providence, 2012.
O. Klopp et al. Oracle inequalities for network models and sparse graphon models. Annals of Statistics, 2017.

For the reasons stated above, I think the paper would benefit from a further revision, and recommend a rejection.

**Justification For Why Not Higher Score:**

Unclear presentation; missing related work

**Justification For Why Not Lower Score:**

N/A

---

### Decision · Program_Chairs · 2024-01-16

Reject